# SLIMMABLE NETWORKS FOR CONTRASTIVE SELF-SUPERVISED LEARNING

## ABSTRACT

Self-supervised learning makes great progress in large model pre-training but suffers in training small models. Previous solutions to this problem mainly rely on knowledge distillation and indeed have a two-stage learning procedure: first train a large teacher model, then distill it to improve the generalization ability of small ones. In this work, we present a new one-stage solution to obtain pre-trained small models without extra teachers: slimmable networks for contrastive self-supervised learning (*SlimCLR*). A slimmable network contains a full network and several weight-sharing sub-networks. We can pre-train for only one time and obtain various networks including small ones with low computation costs. However, in self-supervised cases, the interference between weight-sharing networks leads to severe performance degradation. One evidence of the interference is *gradient imbalance*: a small proportion of parameters produces dominant gradients during backpropagation, and the main parameters may not be fully optimized. The interference between networks also result in *gradient direction divergence*. To overcome these problems, we make the main parameters produce dominant gradients and provide consistent guidance for sub-networks via three techniques: slow start training of sub-networks, online distillation, and loss re-weighting according to model sizes. Besides, a switchable linear probe layer is applied during linear evaluation to avoid the interference of weight-sharing linear layers. We instantiate SlimCLR with typical contrastive learning frameworks and achieve better performance than previous arts with fewer parameters and FLOPs.

## 1 INTRODUCTION

In the past decade, deep learning achieves great success in different fields of artificial intelligence. A large amount of manually labeled data is the fuel behind such success. However, manually labeled data is expensive and far less than unlabeled data in practice. To relieve the constraint of costly annotations, self-supervised learning (Dosovitskiy et al., 2016; Wu et al., 2018; van den Oord et al., 2018; He et al., 2020; Chen et al., 2020a) aims to learn transferable representations for downstream tasks by training networks on unlabeled data. Great progress is made in large models, *i.e.*, models bigger than ResNet-50 (He et al., 2016) that has roughly 25M parameters. For example, ReLICv2 (Tomasev et al., 2022) achieves 77.1% accuracy on ImageNet (Russakovsky et al., 2015) under linear evaluation protocol with ResNet-50, outperforming the supervised baseline 76.5%.

In contrast to the success of the large model pre-training, self-supervised learning with small models lags behind. For instance, supervised ResNet-18 with 12M parameters achieves 72.1% accuracy on ImageNet, but its self-supervised result with MoCov2 (Chen et al., 2020c) is only 52.5% (Fang et al., 2021). The gap is nearly 20%. To fulfill the large performance gap between supervised and self-supervised small models, previous methods (Fang et al., 2021; Gao et al., 2022; Xu et al., 2022) mainly focus on knowledge distillation, namely, they try to transfer the knowledge of a self-supervised large model into small ones. Nevertheless, such methodology actually has a two-stage procedure: first train an additional large model, then train a small model to mimic the large one. Besides, one-time distillation only produces a single small model for a specific computation scenario.

An interesting question naturally arises: can we obtain different small models through one time pre-training to meet various computation scenarios without extra teachers? Inspired by the success of slimmable networks (Yu et al., 2019) in supervised learning, we present a novel one-stage

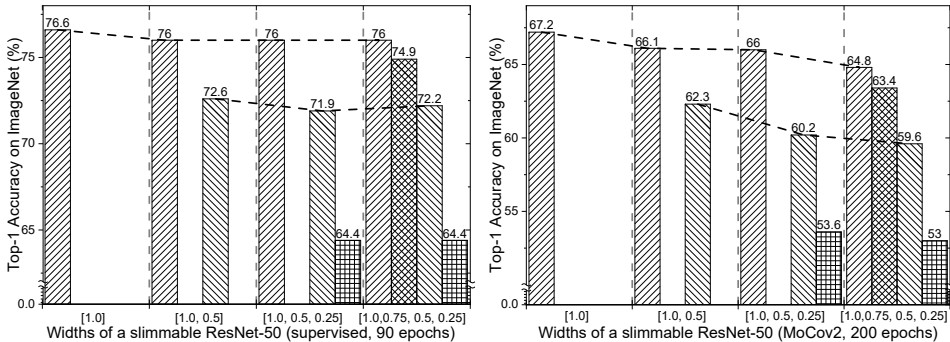

Figure 1: Training a slimmable ResNet-50 in supervised (left) and self-supervised (right) manners. ResNet-50$_{[1.0,0.75,0.5,0.25]}$ means this slimmbale network can switch at width $[1.0, 0.75, 0.5, 0.25]$. The width $0.25$ represents that the number of channels is scaled by $0.25$ of the full network.

method to obtain pre-trained small models without adding large models: slimmable networks for contrastive self-supervised learning (*SlimCLR*). A slimmable network consists of a full network and some weight-sharing sub-networks with different widths. The width denotes the number of channels in a network. Slimmable networks can execute at various widths, permitting flexible deployment on different computing devices. We can thus obtain multiple networks including small ones meeting low computing cases via one-time pre-training. Weight-sharing networks can also inherit knowledge from the large ones via the sharing parameters to achieve better generalization performance.

Weight-sharing networks in a slimmbale network cause interference to each other when training simultaneously, and the situation is worse in self-supervised cases. As shown in Figure 1, with supervision, weight-sharing networks only have a slight impact on each other, *e.g.*, the full model achieves 76.6% *vs.* 76.0% accuracy in ResNet-50$_{[1.0]}$ and ResNet-50$_{[1.0,0.75,0.5,0.25]}$. Without supervision, the corresponding numbers become 67.2% *vs.* 64.8%. One observed phenomenon of the interference is *gradient imbalance*: a small proportion of parameters produces dominant gradients during backpropagation. The imbalance occurs because the sharing parameters receive gradients from multiple losses of different networks during optimization. The main parameters may not be fully optimized due to gradient imbalance. Besides, the conflicts in gradient directions of weight-sharing networks also cause *gradient direction divergence* of the full network. Please refer to Appendix A.3 for detailed explanations and visualizations.

To relieve the gradient imbalance, the main parameters should produce dominant gradients during the optimization process. To avoid conflicts in gradient directions of various networks, sub-networks should have consistent guidance. Following these principles, we introduce three simple yet effective techniques during pre-training to relieve the interference of networks. 1) We adopt a *slow start* strategy for sub-networks. The networks and pseudo supervision of contrastive learning are both unstable and fast updating at the start of training. To avoid interference making the situation worse, we only train the full model at first. After the full model becomes relatively stable, sub-networks can inherit the knowledge via sharing parameters and start with better initialization. 2) We apply *online distillation* to make all sub-networks consistent with the full model to eliminate divergence of networks. The predictions of the full model will serve as global guidance for all sub-networks. 3) We *re-weight the losses of networks* according to their widths to ensure that the full model dominates the optimization process. Besides, we adopt a *switchable linear probe layer* to avoid interference of weight-sharing linear layers during evaluation. A single slimmable linear layer cannot achieve several complex mappings simultaneously when the data distribution is complicated.

We instantiate two algorithms for SlimCLR with typical contrastive learning frameworks, *i.e.*, Mo-Cov2 and MoCov3 (Chen et al., 2020c; 2021). Extensive experiments are done on ImageNet (Russakovsky et al., 2015) dataset, and the results show that our methods achieve significant performance improvements compared to previous arts with fewer parameters and FLOPs.

## 2 RELATED WORKS

**Self-supervised learning**  Self-supervised learning aims to learn transferable representations for downstream tasks from the input data itself. According to Liu et al. (2020), self-supervised methods can be summarized into three main categories according to their objectives: *generative*, *contrastive*,

and *generative-contrastive (adversarial)*. Methods belonging to the same categories can be further classified by the difference between pretext tasks. Given input $x$, generative methods encode $x$ into an explicit vector $z$ and decode $z$ to reconstruct $x$ from $z$, *e.g.*, auto-regressive (van den Oord et al., 2016a;b), auto-encoding models (Ballard, 1987; Kingma & Welling, 2014; Devlin et al., 2019; He et al., 2022). Contrastive learning methods encoder input $x$ into an explicit vector $z$ to measure similarity. The two mainstream methods below this category are context-instance contrast (info-Max Hjelm et al. (2019), CPC van den Oord et al. (2018), AMDIM Bachman et al. (2019)) and instance-instance contrast (DeepCluster Caron et al. (2018), MoCo He et al. (2020); Chen et al. (2021), SimCLR Chen et al. (2020a;b), SimSiam Chen & He (2021)). Generative-contrastive methods generate a fake sample $x'$ from $x$ and try to distinguish $x'$ from real samples, *e.g.*, DCGANs Radford et al. (2016), inpainting Pathak et al. (2016), and colorization Zhang et al. (2016).

**Slimmable networks** Slimmable networks are first proposed to achieve instant and adaptive accuracy-efficiency trade-offs on different devices (Yu et al., 2019). It can execute at different widths during runtime. Following the pioneering work, universally slimmable networks (Yu & Huang, 2019b) develop systematic training approaches to allow slimmable networks to run at arbitrary widths. AutoSlim (Yu & Huang, 2019a) further achieves one-shot architecture search for channel numbers under a certain computation budget. MutualNet (Yang et al., 2020) trains slimmable networks using different input resolutions to learn multi-scale representations. Dynamic slimmable networks (Li et al., 2022; 2021) change the number of channels of each layer in the fly according to the input. In contrast to weight-sharing sub-networks in slimmable networks, some methods try to train multiple sub-networks with independent parameters (Zhao et al., 2022b). A relevant concept of slimmable networks in network pruning is *network slimming* (Liu et al., 2017; Chavan et al., 2022; Wang et al., 2021), which aims to achieve channel-level sparsity for computation efficiency.

## 3 METHOD

### 3.1 DESCRIPTION OF SLIMCLR

We develop two instantial algorithms for SlimCLR with typical contrastive learning frameworks Mo-Cov2 and MoCov3 (Chen et al., 2020c; 2021). As shown in Figure 2a (right), a slimmable network with $n$ widths $w_1, \ldots, w_n$ contains multiple weight-sharing networks $f_{\theta_{w_1}}, \ldots, f_{\theta_{w_n}}$, which are parameterized by learnable weights $\theta_{w_1}, \ldots, \theta_{w_n}$, respectively. Each network $f_{\theta_{w_i}}$ in the slimmable network has its own set of weights $\Theta_{w_i}$ and $\theta_{w_i} \in \Theta_{w_i}$. A network with a small width shares the weights with large ones, namely, $\Theta_{w_j} \subset \Theta_{w_i}$ if $w_j < w_i$. Generally, we assume $w_j < w_i$ if $j > i$, *i.e.*, $w_1, \ldots, w_n$ arrange in descending order, and $\theta_{w_1}$ represent the parameters of the full model.

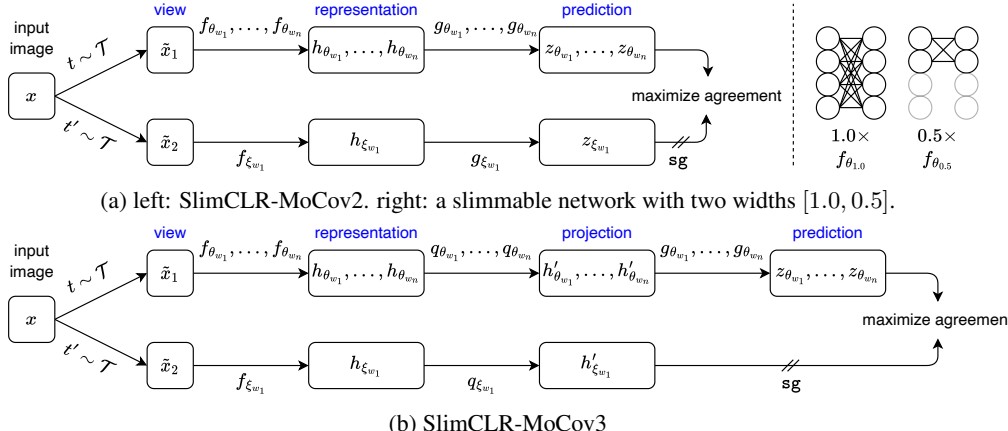

(a) left: SlimCLR-MoCov2. right: a slimmable network with two widths $[1.0, 0.5]$.

(b) SlimCLR-MoCov3

Figure 2: The overall framework of SlimCLR. A slimmable network produces different outputs from weight-sharing networks with various widths $w_1, \ldots, w_n$, where $w_1$ is the width of the full model. $\theta$ are the network parameters and $\xi$ are an exponential moving average of $\theta$. sg means stop-gradient.

We first illustrate the learning process of SlimCLR-MoCov2 in Figure 2a. Given a set of images $\mathcal{D}$, an image $x$ sampled uniformly from $\mathcal{D}$, and one distribution of image augmentation $\mathcal{T}$, SlimCLR produces two data views $\tilde{x}_1 = t(x)$ and $\tilde{x}_2 = t'(x)$ from $x$ by applying augmentations $t \sim \mathcal{T}$ and $t' \sim \mathcal{T}$, respectively. For the first view, SlimCLR outputs multiple representations $h_{\theta_{w_1}}, \ldots, h_{\theta_{w_n}}$ and predictions $z_{\theta_{w_1}}, \ldots, z_{\theta_{w_n}}$, where $h_{\theta_{w_i}} = f_{\theta_{w_i}}(\tilde{x}_1)$ and $z_{\theta_{w_i}} = g_{\theta_{w_i}}(h_{\theta_{w_i}})$. $g$ is a stack of slimmable linear transformation layers, *i.e.*, a slimmable version of the MLP head in MoCov2 and SimCLR (Chen et al., 2020a). For the second view, SlimCLR only outputs a single representation from the full model $h_{\xi_{w_1}} = f_{\xi_{w_1}}(\tilde{x}_2)$ and prediction $z_{\xi_{w_1}} = g_{\xi_{w_1}}(h_{\xi_{w_1}})$. We minimize the InfoNCE (van den Oord et al., 2018) loss to maximize the similarity of positive pairs $z_{\theta_{w_i}}$ and $z_{\xi_{w_1}}$:

$$\mathcal{L}_{z_{\theta_{w_i}}, z_\xi, \{z^-\}} = -\log \frac{\exp(\overline{z}_{\theta_{w_i}} \cdot \overline{z}_{\xi_{w_1}} / \tau_1)}{\exp(\overline{z}_{\theta_{w_i}} \cdot \overline{z}_{\xi_{w_1}} / \tau_1) + \sum_{z^-} \exp(\overline{z}_{\theta_{w_i}} \cdot z^- / \tau_1)}, \tag{1}$$

where $\overline{z}_{\theta_{w_i}} = z_{\theta_{w_i}} / \|z_{\theta_{w_i}}\|_2$, $\overline{z}_{\xi_{w_1}} = z_{\xi_{w_1}} / \|z_{\xi_{w_1}}\|_2$, $\tau_1$ is a temperature hyper-parameter, and $\{z^-\}$ are features of negative samples. For SlimCLR-MoCov2, $\{z^-\}$ comes from a queue. Following MoCov2, the queue is updated by $\overline{z}_{\xi_{w_1}}$ every iteration during training. The overall objective is the sum of losses of all networks with various widths:

$$\mathcal{L}_{z_\theta, z_\xi, \{z^-\}} = \sum_{i=1}^{n} \mathcal{L}_{z_{\theta_{w_i}}, z_\xi, \{z^-\}}. \tag{2}$$

$\xi$ is updated by $\theta$ every iteration: $\xi \leftarrow m\xi + (1-m)\theta$, where $m \in [0, 1)$ is a momemtum coefficient.

Compared to SlimCLR-MoCov2, SlimCLR-MoCov3 has an additional projection process. It first projects the representation to another high dimensional space, then makes predictions. The projector $q$ is a stack of slimmable linear transformation layers. SlimCLR-MoCov3 also adopts the InfoNCE loss, but the negative samples come from other samples in the mini-batch.

After contrastive learning, we only keep $f_{\theta_{w_1}}, \ldots, f_{\theta_{w_n}}$ and abandon other components.

## 3.2 GRADIENT IMBALANCE AND SOLUTIONS

As shown in Figure 1, a vanilla implementation of the above framework leads to severe performance degradation as weight-sharing networks interfere with each other during pre-training. One evidence of such interference we observed is *gradient imbalance*.

*Gradient imbalance* refers to that a small proportion of parameters produces dominant gradients during backpropagation. To quantitatively evaluate the phenomenon, we show the ratios of gradient norms of main and minor parameters: $\|\nabla_{\theta_{1.0}}\mathcal{L}\|_2$ and $\|\nabla_{\theta_{1.0\backslash 0.25}}\mathcal{L}\|_2$ versus $\|\nabla_{\theta_{0.25}}\mathcal{L}\|_2$ in Figure 3, where $\mathcal{L}$ is the loss function. Meanwhile, the ratio of the numbers of parameters is $\frac{|\Theta_{1.0}\backslash\Theta_{0.25}|}{|\Theta_{0.25}|} \approx 15$, where $\theta_{1.0\backslash 0.25} \in \Theta_{1.0}\backslash\Theta_{0.25}$. This means $\Theta_{1.0}\backslash\Theta_{0.25}$ contains more than $90\%$ of the total parameters. Generally, the main parameters dominate the optimization process and produce large gradient norms, *i.e.*, the two ratios should both be large ($> 1$). In Figure 3a, the two ratios are both around 3.5 when training a normal network. However, in Figure 3b and 3c, when training a slimmable network, gradient imbalance occurs because sharing parameters obtain multiple gradients from different losses. To be specific, if the widths of a slimmable network $w_1, \ldots, w_n$ arrange in a descending order and the training loss is $\mathcal{L}_{z_\theta, z_\xi, \{z^-\}}$, $\theta_{w_n}$ that only represent a small part of parameters will receive gradients from $n$ different losses and obtain a large gradient norm:

$$\nabla_{\theta_{w_n}}\mathcal{L}_{z_\theta, z_\xi, \{z^-\}} = \frac{\partial \mathcal{L}_{z_\theta, z_\xi, \{z^-\}}}{\partial \theta_{w_n}} = \sum_{i=1}^{n} \frac{\partial \mathcal{L}_{z_{\theta_{w_i}}, z_\xi, \{z^-\}}}{\partial \theta_{w_n}}. \tag{3}$$

Gradient imbalance is more obvious in self-supervised cases. In the supervised case in Figure 3b, $\|\nabla_{\theta_{1.0\backslash 0.25}}\mathcal{L}\|_2$ is close to $\|\nabla_{\theta_{0.25}}\mathcal{L}\|_2$ at first, and the former becomes larger along with the training process. By contrast, for vanilla SlimCLR-MoCov2 in Figure 3c, $\|\nabla_{\theta_{1.0\backslash 0.25}}\mathcal{L}\|_2$ is smaller than the other most time. A conjecture is that instance discrimination is harder than supervised classification. Consequently, small networks with limited capacity are hard to convergence, produce large losses, and cause more disturbances to other weight-sharing networks.

The gradient directions of weight-sharing networks may also diverge from each other during backpropagation. This causes the *gradient direction divergence* of the full network during training.

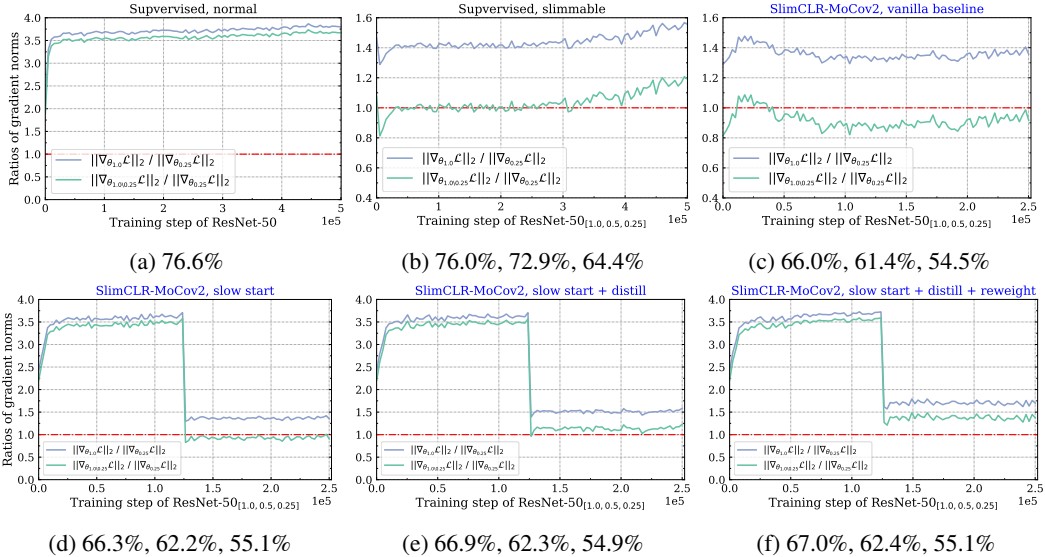

Figure 3: Ratios of gradient norms: $\frac{\|\nabla_{\theta_{1.0}}\mathcal{L}\|_2}{\|\nabla_{\theta_{0.25}}\mathcal{L}\|_2}$ and $\frac{\|\nabla_{\theta_{1.0\backslash 0.25}}\mathcal{L}\|_2}{\|\nabla_{\theta_{0.25}}\mathcal{L}\|_2}$. The gradient norm of the network is calculated by averaging the layer-wise $\ell_2$ gradient norms. $\nabla_{\theta_{1.0\backslash 0.25}}\mathcal{L}$ is the gradient of the final loss *w.r.t.* parameters $\theta_{1.0\backslash 0.25} \in \Theta_{1.0}\backslash\Theta_{0.25}$, *i.e.*, rest parameters of $\Theta_{1.0}$ besides $\Theta_{0.25}$.

Specifically, the gradient directions of the full network are unstable and change fast during training as shown in Figure 5 in Appendix A.3.2.

To avoid gradient imbalance, one natural idea is to make the main parameters dominate the optimization process, *i.e.*, the two ratios in Figure 3 should both be large. To resolve the conflicts of gradient directions, networks should have a consistent optimization goal. In order to achieve the above purposes, we develop three simple yet effective techniques during pre-training: *slow start*, *online distillation*, and *loss reweighting*. Besides, we further introduce a *switchable linear probe layer* to avoid the interference of weight-sharing linear layers during linear evaluation.

**slow start** At the start of training, the model and pseudo supervision of contrastive learning are both fast updating. The optimization procedure is unstable. To avoid interference between weight-sharing networks making the situation harder, at the first $S$ epochs, we only train the full model, *i.e.*, only update $\theta_{1.0}$ by $\nabla_{\theta_{1.0}}\mathcal{L}_{z_{\theta_{1.0}},z_\xi,\{z^-\}}$. In Figure 3d, the ratios of gradient norms are large before the $S$-th epoch; then they dramatically become small after slow start. At the first $S$ epochs, the full model can learn certain knowledge from the data without disturbances, and sub-networks can inherit the knowledge via the sharing parameters and start with better initialization. Similar approaches are also adopted in some one-shot NAS methods (Cai et al., 2020; Yu et al., 2020).

**online distillation** The full model has the greatest capacity to learn knowledge from the data. The prediction of the full model can serve as consistent guidance for all sub-networks to resolve the gradient direction conflicts of weight-sharing networks. Following Yu & Huang (2019b), we minimize the Kullback-Leibler (KL) divergence between the estimated probabilities $p_{w_i} = \frac{\exp(\overline{z}_{\theta_{w_i}}\cdot\overline{z}_{\xi_{w_1}}/\tau_2)}{\exp(\overline{z}_{\theta_{w_i}}\cdot\overline{z}_{\xi_{w_1}}/\tau_2)+\sum_{z^-}\exp(\overline{z}_{\theta_{w_i}}\cdot z^-/\tau_2)}$ of sub-networks and the full model:

$$\mathcal{L}_{p_{w_i}} = -p_{w_1}\log p_{w_i} \quad \text{where } w_i \in \{w_2, \dots, w_n\}. \tag{4}$$

$\tau_2$ is a temperature coefficient of distillation. In Figure 3e, we observe that online distillation helps $\|\nabla_{\theta_{1.0\backslash 0.25}}\mathcal{L}\|_2/\|\nabla_{\theta_{0.25}}\mathcal{L}\|_2$ become larger than 1.0. This means that online distillation also relieves the gradient imbalance and helps the main parameters dominate the optimization process.

**loss reweighting** Another straightforward solution to gradient imbalance and gradient direction divergence is to assign large confidence to networks with large widths. We adopt a strategy in which

the strongest takes control. The weight for the loss of the network with width $w_i$ is:

$$\lambda_i = 1.0 + \mathbf{1}\{w_i = w_1\} \times \sum_{j=2}^{n} w_j,$$ (5)

where $\mathbf{1}\{\cdot\}$ equals to 1 if the inner condition is true, 0 otherwise. In Figure 3f, both ratios become large, and $\|\nabla_{\theta_{1.0\backslash 0.25}}\mathcal{L}\|_2/\|\nabla_{\theta_{0.25}}\mathcal{L}\|_2$ are larger than 1.0 by a clear margin. Loss reweighting helps the main parameters produce large gradient norms and dominate the optimization process.

The overall pre-training objective of SlimCLR is:

$$\mathcal{L}_{all} = \lambda_1 \mathcal{L}_{z_{\theta_{w_1}}, z_\xi, \{z^-\}} + \sum_{i=2}^{n} \lambda_i \frac{(\mathcal{L}_{z_{\theta_{w_i}}, z_\xi, \{z^-\}} + \mathcal{L}_{p_{w_i}})}{2}.$$ (6)

**switchable linear probe layer** As we demonstrate theoretically in Appendix A.1, given the features extracted by a slimmable network which is pre-trained via contrastive self-supervised learning methods, a single slimmable linear probe layer cannot achieve several complex mappings from different representations to the same object classes simultaneously. The failure is because the learned representations in Figure 2 do not meet the requirement discussed in Appendix A.1. In this case, we propose a switchable linear probe layer mechanism. Namely, each network in the slimmable network will have its own linear probe layer for linear evaluation.

## 4 EXPERIMENTS

### 4.1 EXPERIMENTAL DETAILS

**Datatest** We train SlimCLR on ImageNet (Russakovsky et al., 2015), which contains 1.28M training and 50K validation images. During pre-training, we use training images without labels.

**Learning strategies of SlimCLR-MoCov2** By default, we use a total batch size 1024, an initial learning rate 0.2, and weight decay $1 \times 10^{-4}$. We adopt the SGD optimizer with a momentum 0.9. A linear warm-up and cosine decay policy (Goyal et al., 2017; He et al., 2019) for learning rate is applied, and the warm-up epoch is 10. The temperatures are $\tau_1 = 0.2$ for InfoNCE and $\tau_2 = 5.0$ for online distillation. Without special mentions, other settings including data augmentations, queue size (65536), and feature dimension (128) are the same as the counterparts of MoCov2 (Chen et al., 2020c). The slow start epoch $S$ of sub-networks is set to be half of the number of total epochs.

**Learning strategies of SlimCLR-MoCov3** We use a total batch size 1024, an initial learning rate 1.2, and weight decay $1 \times 10^{-6}$. We adopt the LARS (You et al., 2017) optimizer and a cosine learning rate policy with warm-up epoch 10. The temperatures are $\tau_1 = 1.0$ and $\tau_2 = 1.0$. The slow start epoch $S$ is half of the total epochs. One different thing is that we increase the initial learning rate to 3.2 after $S$ epochs. Pre-training is all done with mixed precision (Micikevicius et al., 2018).

**Linear evaluation** Following the general linear evaluation protocol (Chen et al., 2020a; He et al., 2020), we add new linear layers on the backbone and freeze the backbone during evaluation. We also apply online distillation with a temperature $\tau_2 = 1.0$ when training these linear layers. For evaluation of SlimCLR-MoCov2, we use a total batch size 1024, epochs 100, and an initial learning rate 60, which is decayed by 10 at 60 and 80 epochs. For evaluation of SlimCLR-MoCov3, we use a total batch size 1024, epochs 90, and an initial learning rate 0.4 with cosine decay policy.

### 4.2 RESULTS OF SLIMCLR ON IMAGENET

Results of SlimCLR on ImageNet are shown in Table 1. Even though we pay huge efforts to relieve the interference of weight-sharing networks as described in Section 3.2, slimmable training inevitably leads to a drop in performance for the full model. When training for more epochs, the degradation is more obvious. However, such degradation also occurs in the supervised case. Considering the advantages of slimmable training we will discuss below, the degradation is acceptable.

Compared to MoCov2 with individual networks, SlimCLR helps sub-networks achieve significant performance improvements. Specifically, for ResNet-50$_{0.5}$ and ResNet-50$_{0.25}$, SlimCLR-MoCov2

| Method | Backbone | Teacher | Top-1 | Top-5 | Epochs | #Params | #FLOPs |
|---|---|---|---|---|---|---|---|
| Supervised | R-50 | ✗ | 76.6 | 93.2 | 100 | 25.6 M | 4.1 G |
|  | R-34 |  | 75.0 | - | - | 21.8 M | 3.7 G |
|  | R-18 |  | 72.1 | - | - | 11.9 M | 1.8 G |
|  | R-50$_{1.0}$ |  | 76.0$_{(0.6↓)}$ | 92.9 |  | 25.6 M | 4.1 G |
|  | R-50$_{0.75}$ | ✗ | 74.9 | 92.3 | 100 | 14.7 M | 2.3 G |
|  | R-50$_{0.5}$ |  | 72.2 | 90.8 |  | 6.9 M | 1.1 G |
|  | R-50$_{0.25}$ |  | 64.4 | 86.0 |  | 2.0 M | 278 M |
| Baseline (individual networks trained with MoCov2) | R-50 | ✗ | 67.5 | - | 200 | 25.6 M | 4.1 G |
|  | R-50$_{1.0}$ |  | 67.2 | 87.8 |  | 25.6 M | 4.1 G |
|  | R-50$_{0.75}$ | ✗ | 64.3 | 85.8 | 200 | 14.7 M | 2.3 G |
|  | R-50$_{0.5}$ |  | 58.9 | 82.2 |  | 6.9 M | 1.1 G |
|  | R-50$_{0.25}$ |  | 47.9 | 72.8 |  | 2.0 M | 278 M |
| MoCov2 (2020c, preprint) | R-50 |  | 71.1 | - | 800 |  |  |
| MoCov3 (2021, ICCV) | R-50 |  | 72.8 | - | 300 |  |  |
| SlimCLR-MoCov2 | R-50$_{1.0}$ | ✗ | 67.4$_{(0.1↓)}$ | 87.9 | 200 | 25.6 M | 4.1 G |
| SlimCLR-MoCov2 | R-50$_{1.0}$ |  | 70.4$_{(0.7↓)}$ | 89.6 | 800 |  |  |
| SlimCLR-MoCov3 | R-50$_{1.0}$ |  | 72.3$_{(0.5↓)}$ | 90.8 | 300 |  |  |
| SEED (2021, ICLR) | R-34 | R-50 (67.4) | 58.5 | 82.6 | 200 |  |  |
| DisCo (2022, ECCV) | R-34 | R-50 (67.4) | 62.5 | 85.4 | 200 |  |  |
| BINGO (2022, ICLR) | R-34 | R-50 (67.4) | 63.5 | 85.7 | 200 | 21.8 M | 3.7 G |
| SEED (2021, ICLR) | R-34 | R-50×2 (77.3) | 65.7 | 86.8 | 800 |  |  |
| DisCo (2022, ECCV) | R-34 | R-50×2 (77.3) | 67.6 | 88.6 | 200 |  |  |
| BINGO (2022, ICLR) | R-34 | R-50×2 (77.3) | 68.9$_{(6.1)}$ | 88.9 | 200 |  |  |
| SlimCLR-MoCov2 | R-50$_{0.75}$ |  | 65.5 | 87.0 | 200 |  |  |
| SlimCLR-MoCov2 | R-50$_{0.75}$ | ✗ | 68.8 | 88.8 | 800 | 14.7 M | 2.3 G |
| SlimCLR-MoCov3 | R-50$_{0.75}$ |  | **69.7$_{(5.2)}$** | **89.4** | 300 |  |  |
| CompRess (2020, NeurIPS) | R-18 | R-50 (71.1) | 62.6 | - | 130 |  |  |
| SEED (2021, ICLR) | R-18 | R-50×2 (77.3) | 63.0 | 84.9 | 800 |  |  |
| DisCo (2022, ECCV) | R-18 | R-50×2 (77.3) | 65.2 | 86.8 | 200 |  |  |
| BINGO (2022, ICLR) | R-18 | R-50×2 (77.3) | 65.5 | 87.0 | 200 | 11.9 M | 1.8 G |
| SEED (2021, ICLR) | R-18 | R-152 (74.1) | 59.5 | 65.5 | 200 |  |  |
| DisCo (2022, ECCV) | R-18 | R-152 (74.1) | 65.5 | 86.7 | 200 |  |  |
| BINGO (2022, ICLR) | R-18 | R-152 (74.1) | 65.9$_{(6.2)}$ | 87.1 | 200 |  |  |
| SlimCLR-MoCov2 | R-50$_{0.5}$ |  | 62.5 | 84.8 | 200 |  |  |
| SlimCLR-MoCov2 | R-50$_{0.5}$ | ✗ | 65.6 | 87.2 | 800 | 6.9 M | 1.1 G |
| SlimCLR-MoCov3 | R-50$_{0.5}$ |  | **67.6$_{(4.6)}$** | **88.2** | 300 |  |  |
| SlimCLR-MoCov2 | R-50$_{0.25}$ |  | 55.1 | 79.5 | 200 |  |  |
| SlimCLR-MoCov2 | R-50$_{0.25}$ | ✗ | 57.6 | 81.5 | 800 | 2.0 M | 278 M |
| SlimCLR-MoCov3 | R-50$_{0.25}$ |  | **62.4$_{(2.0)}$** | **84.4** | 300 |  |  |

Table 1: Linear evaluation results of SlimCLR with ResNet-50$_{[1.0,0.75,0.5.0.25]}$ on ImageNet. Through only one-time pre-training, SlimCLR obtains multiple different small models without extra large teacher models. It also outperforms previous methods using ResNet as backbones. The performance degradation when training a slimmable network is shown in cyan. The gaps between the self-supervised and supervised results are shown in orange. The smaller, the better.

achieves 3.5% and 6.6% improvements in performance when pre-training for 200 epochs, respectively. This verifies that sub-networks can inherit knowledge from the full model via sharing parameters to improve their generalization ability. We can also use more powerful contrastive learning framework to further boost the performance of sub-networks, *i.e.*, SlimCLR-MoCov3.

Compared to previous methods that aim to distill the knowledge of large teacher models, sub-networks of ResNet-50$_{[1.0,0.75,0.5.0.25]}$ achieve better performance with fewer parameters and FLOPs. SlimCLR also helps small models get closer performance to their supervised counterparts. Furthermore, SlimCLR does not need any additional training process of large teacher models, and all networks in SlimCLR are trained jointly. By only training for one time, we can get different models with various computation cost which are suitable for different devices. This demonstrates the superiority of adopting slimmable networks for contrastive learning to get pre-trained small models.

## 4.3 DISCUSSION

In this section, we will discuss the influences of different components in SlimCLR.

| Model | slimmable Top-1 | Top-5 | switchable Top-1 | Top-5 |
|---|---|---|---|---|
| $R\text{-}50_{1.0}$ | 64.8 | 86.1 | **65.6** | **86.8** |
| $R\text{-}50_{0.75}$ | 63.4 | 85.3 | **64.3** | **86.0** |
| $R\text{-}50_{0.5}$ | 59.6 | 82.9 | **61.3** | **84.1** |
| $R\text{-}50_{0.25}$ | 53.0 | 77.8 | **54.5** | **79.1** |

(a) switchable linear probe layer

| Model | $S=0$ Top-1 | Top-5 | $S=100$ Top-1 | Top-5 |
|---|---|---|---|---|
| $R\text{-}50_{1.0}$ | 65.6 | 86.8 | **66.7** | **87.5** |
| $R\text{-}50_{0.75}$ | 64.3 | 86.0 | **65.3** | **86.4** |
| $R\text{-}50_{0.5}$ | 61.3 | 84.1 | **62.5** | **84.3** |
| $R\text{-}50_{0.25}$ | 54.5 | 79.1 | **54.9** | **79.5** |

(b) slow start epoch $S$, 200 epochs

| Model | MSE | ATKD | DKD Top-1 | KD |
|---|---|---|---|---|
| $R\text{-}50_{1.0}$ | 66.9 | 66.4 | 66.8 | **67.0** |
| $R\text{-}50_{0.75}$ | 65.2 | 65.0 | 65.1 | **65.3** |
| $R\text{-}50_{0.5}$ | 62.4 | 62.2 | 62.3 | **62.6** |
| $R\text{-}50_{0.25}$ | **55.2** | 54.7 | 54.9 | 54.9 |

(c) online distillation loss choice

| Model | 3.0 | 4.0 | 5.0 Top-1 | 6.0 |
|---|---|---|---|---|
| $R\text{-}50_{1.0}$ | **67.0** | **67.0** | **67.0** | 66.7 |
| $R\text{-}50_{0.75}$ | 65.2 | **65.3** | **65.3** | 65.2 |
| $R\text{-}50_{0.5}$ | 62.4 | **62.6** | **62.6** | 62.4 |
| $R\text{-}50_{0.25}$ | **55.0** | 54.8 | 54.9 | **55.0** |

(d) online distillation temperature

| Model | (1) | (2) | (3) Top-1 | (4) |
|---|---|---|---|---|
| $R\text{-}50_{1.0}$ | 67.4 | 67.3 | **67.5** | **67.5** |
| $R\text{-}50_{0.75}$ | 65.5 | 65.7 | **65.9** | 65.8 |
| $R\text{-}50_{0.5}$ | 62.5 | 62.2 | **62.6** | 62.4 |
| $R\text{-}50_{0.25}$ | **55.1** | 54.5 | 54.4 | 54.5 |

(e) loss reweighting

| Model | 200 | 300 | 400 Top-1 | 500 |
|---|---|---|---|---|
| $R\text{-}50_{1.0}$ | 70.2 | 70.0 | **70.3** | 70.1 |
| $R\text{-}50_{0.75}$ | 68.3 | 68.6 | **68.8** | 68.4 |
| $R\text{-}50_{0.5}$ | **65.7** | 65.6 | 65.6 | 65.3 |
| $R\text{-}50_{0.25}$ | **57.8** | 57.5 | 57.6 | 57.3 |

(f) slow start epoch $S$, 800 epochs

Table 2: Ablation experiments with SlimCLR-MoCov2 on ImageNet. The experiment in a former table serves as a baseline for the consequent table.

**switchable linear probe layer** The influence of the switchable linear probe layer is shown in Table 2a. A switchable linear probe layer brings significant improvements in accuracy compared to slimmable linear probe layer. For only one slimmable layer, the interference between weight-sharing linear layers is not unavoidable. It is also possible that the learned representations of pre-trained models do not meet the requirements as we discussed in Appendix A.1.

**slow start and training time** Experiments with and without slow start are shown in Table 2b. The pre-training time of SlimCLR-MoCov2 without and with slow start epoch $S = 100$ on 8 Tesla V100 GPUs are 45 and 33 hours, respectively. For reference, the pre-training time of MoCov2 with ResNet-50 is roughly 20 hours. Slow start largely reduces the pre-training time. It also avoids the interference between weight-sharing networks at the start stage of training and helps the system reach a steady point fast during optimization. Thus sub-networks can start with good initialization and achieve better performance. We also provide ablations of slow start epoch $S$ when training for a longer time in Table 2f. Setting $S$ to be half of the total epochs is a natural and appropriate choice.

**online distillation** Here we compare two classical distillation losses: mean-square-error (MSE) and KL divergence (KD), and two other distillation losses from recent works: ATKD (Guo, 2022) and DKD (Zhao et al., 2022a). ATKD reduces the difference in sharpness between distributions of teacher and student to help the student better mimic the teacher. DKD decouples the classical knowledge distillation objective function (Hinton et al., 2015) into target class and non-target class knowledge distillation to achieve more effective and flexible distillation. In Table 2c, we can see that these four distillation losses make trivial differences in our context.

Combining the results of distillation with ResNet$_{[1.0,0.5,0.25]}$ in Figure 3e, we find that distillation mainly improves the performance of the full model, and the improvements of sub-networks are relatively small. This violates the purpose of knowledge distillation: distill the knowledge of large models and improve the performance of small ones. This is possibly because sub-networks in a slimmable network already inherit the knowledge from the full model via the sharing weights, and feature distillation cannot help the sub-networks much in this case. The main function of online distillation in our context is to relieve the interference of sub-networks as shown in Figure 3e.

We also test the influence of different temperatures in online distillation, *i.e.*, $\tau_2$ related to $p_{w_i}$ in equation 4, and results are shown in Table 2d. Following classical KD (Hinton et al., 2015), we choose $\tau_2 \in \{3.0, 4.0, 5.0, 6.0\}$. The choices of temperatures make a trivial difference in our context. SEED (Fang et al., 2021) use a small temperature 0.01 for the teacher to get sharp distribution and a temperature 0.2 for the student. BINGO (Xu et al., 2022) adopts a single temperature 0.2. Their choices are quite different from ours, and SlimCLR is more robust to the choice of temperatures. We further provide an analysis of the influence of temperatures in Appendix A.2.

**loss reweighting** We compared four loss reweighting manners in Table 2e. They are

(1). $\lambda_i = 1.0 + \mathbf{1}\{w_i = w_1\} \times \sum_{j=2}^{n} w_j$,    (2). $\lambda_i = 1.0 + \mathbf{1}\{w_i = w_1\} \times \max\{w_2, \ldots, w_n\}$,

(3). $\lambda_i = n \times \frac{w_i}{\sum_{j=1}^{n} w_j}$,                     (4). $\lambda_i = n \times \frac{1.0 + \sum_{j=i}^{n} w_j}{\sum_{j=1}^{n}(1.0 + \sum_{k=j}^{n} w_k)}$,

| pre-train | backbone | $AP^{bb}$ | $AP^{bb}_{50}$ | $AP^{bb}_{75}$ | $AP^{mk}$ | $AP^{mk}_{50}$ | $AP^{mk}_{75}$ |
|---|---|---|---|---|---|---|---|
| supervised (Yu et al., 2019) | $R\text{-}50_{1.0}$ | 37.4 | 59.6 | 40.5 | 34.9 | 56.5 | 37.3 |
| | $R\text{-}50_{0.75}$ | 36.7 | 58.7 | 39.3 | 34.3 | 55.4 | 36.1 |
| | $R\text{-}50_{0.5}$ | 34.7 | 56.3 | 36.8 | 32.6 | 53.1 | 34.1 |
| | $R\text{-}50_{0.25}$ | 30.2 | 50.3 | 31.5 | 28.6 | 47.5 | 29.9 |
| MoCo (He et al., 2020) | R-50 | 38.5 | 58.9 | 42.0 | 35.1 | 55.9 | 37.7 |
| SEED (Fang et al., 2021) | R-34 | 38.4 | 57.0 | 41.0 | 33.3 | 53.6 | 35.4 |
| BINGO (Xu et al., 2022) | R-18 | 32.0 | 51.0 | 34.7 | 29.6 | 48.2 | 31.5 |
| SlimCLR-MoCov2 | $R\text{-}50_{1.0}$ | 38.2 | 59.8 | 41.6 | **35.1** | 56.5 | 37.5 |
| | $R\text{-}50_{0.75}$ | 37.3 | 58.4 | 40.4 | **34.4** | 55.5 | 36.8 |
| | $R\text{-}50_{0.5}$ | **35.0** | 56.0 | 37.6 | **32.6** | 53.0 | 34.7 |
| | $R\text{-}50_{0.25}$ | **30.7** | 50.3 | 32.6 | **28.7** | 47.5 | 30.5 |

Table 3: Transfer learning results on COCO `val2017` set. Bounding-box AP ($AP^{bb}$) for **object detection** and mask AP ($AP^{mk}$) for **instance segmentation**.

where $\mathbf{1}\{\cdot\}$ equals to 1 if the inner condition is true, 0 otherwise. The corresponding weights of ResNet-$50_{[1.0, 0.75, 0.5, 0.25]}$ are $[2.5, 1.0, 1.0, 1.0]$, $[1.75, 1.0, 1.0, 1.0]$, $[1.6, 1.2, 0.8, 0.4]$, and $[1.54, 1.08, 0.77, 0.62]$. It is clear that a larger weight for the full model helps the system achieve better performance. This demonstrates again that it is important for the full model to lead the optimization direction during training. The differences of the above four loss reweighting strategies are mainly reflected in the sub-networks with small sizes. To ensure the performance of the smallest network, we adopt the reweighting manner (1) in practice.

**Transfer learning to object detection and instance segmentation** Following previous works (He et al., 2020; Fang et al., 2021; Xu et al., 2022), we evaluate the generalization ability of SlimCLR on object detection and instance segmentation tasks. As the supervised slimmable networks (Yu et al., 2019), we use Mask R-CNN (He et al., 2017) with FPN (Lin et al., 2017) for the two tasks. We fine-tune parameters of all layers including batch normalization (Ioffe & Szegedy, 2015) end-to-end on COCO 2017 (Lin et al., 2014) dataset. The training schedule is the default $1\times$ in Chen et al. (2019). The pre-trained backbone we used here is ResNet-$50_{[1.0, 0.75, 0.5, 0.25]}$ pre-trained via SlimCLR-MoCov2 (800 epochs). The transfer learning results are shown in Table 3. SlimCLR-MoCov2 achieves better transfer learning results than the supervised baseline.

**Differences of training slimmable networks in self-supervised and supervised cases** In Appendix A.3, through extensive visualizations, we show that optimization process of training slimmable networks is harder in self-supervised cases than that in supervised cases. Specifically, the gradient imbalance and gradient direction divergence are both more significant in self-supervised cases as we discussed in Section 3.2 and Appendix A.3.2. In supervised cases, clear global supervision can help slimmable networks avoid these problems to some extent during training. In self-supervised learning, we do not have such clear global supervision, and we need to pay more effort to deal with the performance degradation problems.

## 5 CONCLUSION

In this work, we adapt slimmable networks for contrastive learning to obtain pre-trained small models in a self-supervised manner. By using slimmable networks, we can pre-train for one time and get several models with different sizes which are suitable for various devices with different computation resources. Besides, unlike previous distillation based methods, our methods do not require additional training process of large teacher models. However, weight-sharing sub-networks in a slimmable network cause severe interference to each other in self-supervised learning. One evidence of such interference we observed is the gradient imbalance in the backpropagation process. We develop several techniques to relieve the interference of weight-sharing networks during pre-training and linear evaluation. Two specific algorithms are instantiated in this work, *i.e.*, SlimCLR-MoCov2 and SlimCLR-MoCov3. We take extensive experiments on ImageNet and achieve better performance than previous arts with fewer network parameters and FLOPs.

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

# A APPENDIX

## A.1 CONDITIONS OF INPUTS GIVEN A SLIMMABLE LINEAR LAYER

We consider the conditions of inputs when only using one slimmable linear transformation layer, *i.e.*, consider solving multiple multi-class linear regression problems with shared weights. The parameters of the linear layer are $\boldsymbol{\theta} \in \mathbb{R}^{d \times C}$, $C$ is the number of classes, where $\boldsymbol{\theta} = \begin{bmatrix} \boldsymbol{\theta}_{11} \\ \boldsymbol{\theta}_{21} \end{bmatrix}$, $\boldsymbol{\theta}_{11} \in \mathbb{R}^{d_1 \times C}$, $\boldsymbol{\theta}_{21} \in \mathbb{R}^{d_2 \times C}$, $d_1 + d_2 = d$.

The first input for the full model is $\boldsymbol{X} \in \mathbb{R}^{N \times d}$, where $N$ is the number of samples, $\boldsymbol{X} = [\boldsymbol{X}_{11} \quad \boldsymbol{X}_{12}]$, $\boldsymbol{X}_{11} \in \mathbb{R}^{N \times d_1}$, $\boldsymbol{X}_{12} \in \mathbb{R}^{N \times d_2}$. The second input $\boldsymbol{X}_1 \in \mathbb{R}^{N \times d_1}$ is the input feature for the sub-model parameterized by $\boldsymbol{\theta}_{11}$.

Generally, we have $N \geq d > d_1$. We assume that both $\boldsymbol{X}$ and $\boldsymbol{X}_1$ have independent columns, *i.e.*, $\boldsymbol{X}^T \boldsymbol{X}$ and $\boldsymbol{X}_1^T \boldsymbol{X}_1$ are invertible. The ground truth is $\boldsymbol{T} \in \mathbb{R}^{N \times C}$.

The prediction of the full model is $\boldsymbol{Y} = \boldsymbol{X} \boldsymbol{\theta}$, to minimize the sum-of-least-squares loss between prediction and ground truth, we get

$$\boldsymbol{\theta} = \arg\min_{\boldsymbol{\theta}} \|\boldsymbol{X}\boldsymbol{\theta} - \boldsymbol{T}\|_2^2. \tag{7}$$

By setting the derivative *w.r.t.* $\boldsymbol{\theta}$ to 0, we get

$$\boldsymbol{\theta} = \left(\boldsymbol{X}^T \boldsymbol{X}\right)^{-1} \boldsymbol{X}^T \boldsymbol{T}. \tag{8}$$

In the same way, we can get

$$\boldsymbol{\theta}_{11} = \left(\boldsymbol{X}_1^T \boldsymbol{X}_1\right)^{-1} \boldsymbol{X}_1^T \boldsymbol{T}. \tag{9}$$

For $\boldsymbol{X}^T \boldsymbol{X}$, we have

$$\boldsymbol{X}^T \boldsymbol{X} = [\boldsymbol{X}_{11} \quad \boldsymbol{X}_{12}]^T [\boldsymbol{X}_{11} \quad \boldsymbol{X}_{12}] = \begin{bmatrix} \boldsymbol{X}_{11}^T \boldsymbol{X}_{11} & \boldsymbol{X}_{11}^T \boldsymbol{X}_{12} \\ \boldsymbol{X}_{12}^T \boldsymbol{X}_{11} & \boldsymbol{X}_{12}^T \boldsymbol{X}_{12} \end{bmatrix}. \tag{10}$$

We denote the inverse of $\boldsymbol{X}^T \boldsymbol{X}$ is $\boldsymbol{B} = \begin{bmatrix} \boldsymbol{B}_{11} & \boldsymbol{B}_{12} \\ \boldsymbol{B}_{21} & \boldsymbol{B}_{22} \end{bmatrix}$, $\boldsymbol{X}^T \boldsymbol{X} \boldsymbol{B} = \boldsymbol{I}$, as $\boldsymbol{X}^T \boldsymbol{X}$ is a symmetric matrix, its inverse is also symmetric, so $\boldsymbol{B}_{12} = \boldsymbol{B}_{21}^T$. For $\boldsymbol{X}^T \boldsymbol{X} \boldsymbol{B} = \boldsymbol{I}$, we have

$$\boldsymbol{X}^T \boldsymbol{X} \boldsymbol{B} = \begin{bmatrix} \boldsymbol{X}_{11}^T \boldsymbol{X}_{11} & \boldsymbol{X}_{11}^T \boldsymbol{X}_{12} \\ \boldsymbol{X}_{12}^T \boldsymbol{X}_{11} & \boldsymbol{X}_{12}^T \boldsymbol{X}_{12} \end{bmatrix} \begin{bmatrix} \boldsymbol{B}_{11} & \boldsymbol{B}_{12} \\ \boldsymbol{B}_{21} & \boldsymbol{B}_{22} \end{bmatrix} = \begin{bmatrix} \boldsymbol{I}_{d_1} & \boldsymbol{0}_{d_1,\,d_2} \\ \boldsymbol{0}_{d_2,\,d_1} & \boldsymbol{I}_{d_2} \end{bmatrix}. \tag{11}$$

Then we can get

$$\boldsymbol{X}_{11}^T \boldsymbol{X}_{11} \boldsymbol{B}_{11} + \boldsymbol{X}_{11}^T \boldsymbol{X}_{12} \boldsymbol{B}_{21} = \boldsymbol{I}_{d_1}, \tag{12}$$

$$\boldsymbol{X}_{11}^T \boldsymbol{X}_{11} \boldsymbol{B}_{12} + \boldsymbol{X}_{11}^T \boldsymbol{X}_{12} \boldsymbol{B}_{22} = \boldsymbol{0}_{d_1,\,d_2}, \tag{13}$$

$$\boldsymbol{X}_{12}^T \boldsymbol{X}_{11} \boldsymbol{B}_{11} + \boldsymbol{X}_{12}^T \boldsymbol{X}_{12} \boldsymbol{B}_{21} = \boldsymbol{0}_{d_1,\,d_2}, \tag{14}$$

$$\boldsymbol{X}_{12}^T \boldsymbol{X}_{11} \boldsymbol{B}_{12} + \boldsymbol{X}_{12}^T \boldsymbol{X}_{12} \boldsymbol{B}_{22} = \boldsymbol{I}_{d_2}. \tag{15}$$

At the same time

$$\begin{aligned} \boldsymbol{\theta} &= \left(\boldsymbol{X}^T \boldsymbol{X}\right)^{-1} \boldsymbol{X}^T \boldsymbol{T} = \boldsymbol{B} \boldsymbol{X}^T \boldsymbol{T} \\ &= \begin{bmatrix} \boldsymbol{B}_{11} & \boldsymbol{B}_{12} \\ \boldsymbol{B}_{21} & \boldsymbol{B}_{22} \end{bmatrix} [\boldsymbol{X}_{11} \quad \boldsymbol{X}_{12}]^T \boldsymbol{T} \\ &= \begin{bmatrix} \boldsymbol{B}_{11} \boldsymbol{X}_{11}^T + \boldsymbol{B}_{12} \boldsymbol{X}_{12}^T \\ \boldsymbol{B}_{21} \boldsymbol{X}_{11}^T + \boldsymbol{B}_{22} \boldsymbol{X}_{12}^T \end{bmatrix} \boldsymbol{T}, \end{aligned} \tag{16}$$

and

$$\boldsymbol{\theta}_{11} \;=\; \left(\boldsymbol{B}_{11}\boldsymbol{X}_{11}^T + \boldsymbol{B}_{12}\boldsymbol{X}_{12}^T\right)\boldsymbol{T} = \left(\boldsymbol{X}_1^T\boldsymbol{X}_1\right)^{-1}\boldsymbol{X}_1^T\boldsymbol{T}. \tag{17}$$

From equation 14, we get

$$\boldsymbol{B}_{21} = -\left(\boldsymbol{X}_{12}^T\boldsymbol{X}_{12}\right)^{-1}\boldsymbol{X}_{12}^T\boldsymbol{X}_{11}\boldsymbol{B}_{11}, \tag{18}$$

$$\boldsymbol{B}_{12} = -\boldsymbol{B}_{11}\boldsymbol{X}_{11}^T\boldsymbol{X}_{12}\left(\boldsymbol{X}_{12}^T\boldsymbol{X}_{12}\right)^{-1}. \tag{19}$$

Substitute equation 18 into equation 12, we get

$$\boldsymbol{X}_{11}^T\boldsymbol{X}_{11}\boldsymbol{B}_{11} - \boldsymbol{X}_{11}^T\boldsymbol{X}_{12}\left(\boldsymbol{X}_{12}^T\boldsymbol{X}_{12}\right)^{-1}\boldsymbol{X}_{12}^T\boldsymbol{X}_{11}\boldsymbol{B}_{11} = \boldsymbol{I}_{d_1}, \tag{20}$$

$$\boldsymbol{B}_{11} \;=\; \left(\boldsymbol{X}_{11}^T\boldsymbol{X}_{11} - \boldsymbol{X}_{11}^T\boldsymbol{X}_{12}\left(\boldsymbol{X}_{12}^T\boldsymbol{X}_{12}\right)^{-1}\boldsymbol{X}_{12}^T\boldsymbol{X}_{11}\right)^{-1}. \tag{21}$$

At the same time

$$\begin{aligned}
\boldsymbol{\theta}_{11} &= \left(\boldsymbol{B}_{11}\boldsymbol{X}_{11}^T + \boldsymbol{B}_{12}\boldsymbol{X}_{12}^T\right)\boldsymbol{T} \\
&= \boldsymbol{B}_{11}\left(\boldsymbol{X}_{11}^T - \boldsymbol{X}_{11}^T\boldsymbol{X}_{12}\left(\boldsymbol{X}_{12}^T\boldsymbol{X}_{12}\right)^{-1}\boldsymbol{X}_{12}^T\right)\boldsymbol{T} \\
&= \left(\boldsymbol{X}_{11}^T\boldsymbol{X}_{11} - \boldsymbol{X}_{11}^T\boldsymbol{X}_{12}\left(\boldsymbol{X}_{12}^T\boldsymbol{X}_{12}\right)^{-1}\boldsymbol{X}_{12}^T\boldsymbol{X}_{11}\right)^{-1}\left(\boldsymbol{X}_{11}^T - \boldsymbol{X}_{11}^T\boldsymbol{X}_{12}\left(\boldsymbol{X}_{12}^T\boldsymbol{X}_{12}\right)^{-1}\boldsymbol{X}_{12}^T\right)\boldsymbol{T}.
\end{aligned} \tag{22}$$

Combining equation 17 and equation 22, we get the condition of the input

$$\left(\boldsymbol{X}_{11}^T\boldsymbol{X}_{11} - \boldsymbol{X}_{11}^T\boldsymbol{X}_{12}\left(\boldsymbol{X}_{12}^T\boldsymbol{X}_{12}\right)^{-1}\boldsymbol{X}_{12}^T\boldsymbol{X}_{11}\right)^{-1}\left(\boldsymbol{X}_{11}^T - \boldsymbol{X}_{11}^T\boldsymbol{X}_{12}\left(\boldsymbol{X}_{12}^T\boldsymbol{X}_{12}\right)^{-1}\boldsymbol{X}_{12}^T\right)\boldsymbol{T}$$
$$= \left(\boldsymbol{X}_1^T\boldsymbol{X}_1\right)^{-1}\boldsymbol{X}_1^T\boldsymbol{T}. \tag{23}$$

To verify whether equation 23 meets in practice, we sample 2048 images from the training set of ImageNet and use a ResNet-50$_{[1.0,0.75,0.5,0.25]}$ pre-trained by SlimCLR-MoCov2 (800 epochs) to extract the features of these images. The features from ResNet-50$_{1.0}$ denote $\boldsymbol{X} \in \mathbb{R}^{2048\times 1024}$ and features from ResNet-50$_{0.5}$ denote $\boldsymbol{X}_1 \in \mathbb{R}^{2048\times 512}$. We use $\boldsymbol{L}$ to represent the left side of equation 23 and $\boldsymbol{R}$ for the right side. Then we use the extracted features to get the absolute difference between $\boldsymbol{L}$ and $\boldsymbol{R}$, *i.e.*, $|\boldsymbol{L} - \boldsymbol{R}|$. The average value of entries in $|\boldsymbol{L} - \boldsymbol{R}|$ is 1.07. This means a total difference 1096665.50. Similar experiments are performed on the validation set of ImageNet. The average value of entries in $|\boldsymbol{L} - \boldsymbol{R}|$ is 0.88. This means a total difference 903094.19.

**These results demonstrate that features of slimmable networks learned by contrastive self-supervised learning cannot meet the input conditions (equation 23) when using a single slimmable linear probe layer.** This explains why using a switchable linear probe layer achieves much better performance than a single slimmable linear probe layer in Table 2a.

## A.2 INFLUENCE OF TEMPERATURES DURING DISTILLATION

In this section, we will analyze the influence of temperatures when applying distillation. One of the previous methods SEED (Fang et al., 2021) uses different temperatures for the student and teacher, without loss of generalization, we will also adopt a such strategy in our analysis. Specifically, we adopt $\tau_t$ for teacher and $\tau_s$ for student.

The predicted probability for a certain category $i$ of the student is $q_i = \frac{e^{z_i / \tau_s}}{\sum_j e^{z_j / \tau_s}}$, where $z$ is the output of the student model, *i.e.*, logit of the model. The probability for a certain category $i$ of the teacher is $p_i = \frac{e^{v_i / \tau_t}}{\sum_j e^{v_j / \tau_t}}$, where $v$ is the output of the student model. The loss is the KL divergence:

$$\mathcal{L} = -\sum_k p_k \log q_k. \tag{24}$$

The gradient of $q$ *w.r.t.* $z$ is:

$$\frac{\partial q_i}{\partial z_i} = \frac{\frac{1}{\tau_s} e^{z_i / \tau_s} \sum_j e^{z_j / \tau_s} - \frac{1}{\tau_s} e^{z_i / \tau_s} e^{z_i / \tau_s}}{\left(\sum_j e^{z_j / \tau_s}\right)^2} = \frac{1}{\tau_s}\left(q_i - q_i^2\right), \tag{25}$$

$$\frac{\partial q_t}{\partial z_i}|_{t\neq i} = \frac{0 - \frac{1}{\tau_s} e^{z_t / \tau_s} e^{z_i / \tau_s}}{\left(\sum_j e^{z_j / \tau_s}\right)^2} = -\frac{1}{\tau_s} q_i q_t. \tag{26}$$

Similarly,

$$\frac{\partial p_i}{\partial v_i} = \frac{1}{\tau_t}\left(p_i - p_i^2\right), \tag{27}$$

$$\frac{\partial p_t}{\partial v_i} = -\frac{1}{\tau_t} p_i p_t. \tag{28}$$

The gradient of $\mathcal{L}$ *w.r.t.* $z$ is:

$$\frac{\partial \mathcal{L}}{\partial z_i} = -\frac{p_i}{q_i}\frac{\partial q_i}{\partial z_i} + \sum_{t\neq i} -\frac{p_t}{q_t}\frac{\partial q_t}{\partial z_i} \tag{29}$$

$$= \frac{1}{\tau_s}(q_i - p_i) \tag{30}$$

$$= \frac{1}{\tau_s}\left(\frac{e^{z_i / \tau_s}}{\sum_j e^{z_j / \tau_s}} - \frac{e^{v_i / \tau_t}}{\sum_j e^{v_j / \tau_t}}\right). \tag{31}$$

Following classical KD (Hinton et al., 2015), we assume temperatures are much larger than logits and use the first-order Taylor series to approximate the exponential function:

$$\frac{\partial \mathcal{L}}{\partial z_i} \approx \frac{1}{\tau_s}\left(\frac{1 + \frac{z_i}{\tau_s}}{C + \sum_j \frac{z_j}{t_s}} - \frac{1 + \frac{v_i}{\tau_t}}{C + \sum_j \frac{v_j}{t_t}}\right), \tag{32}$$

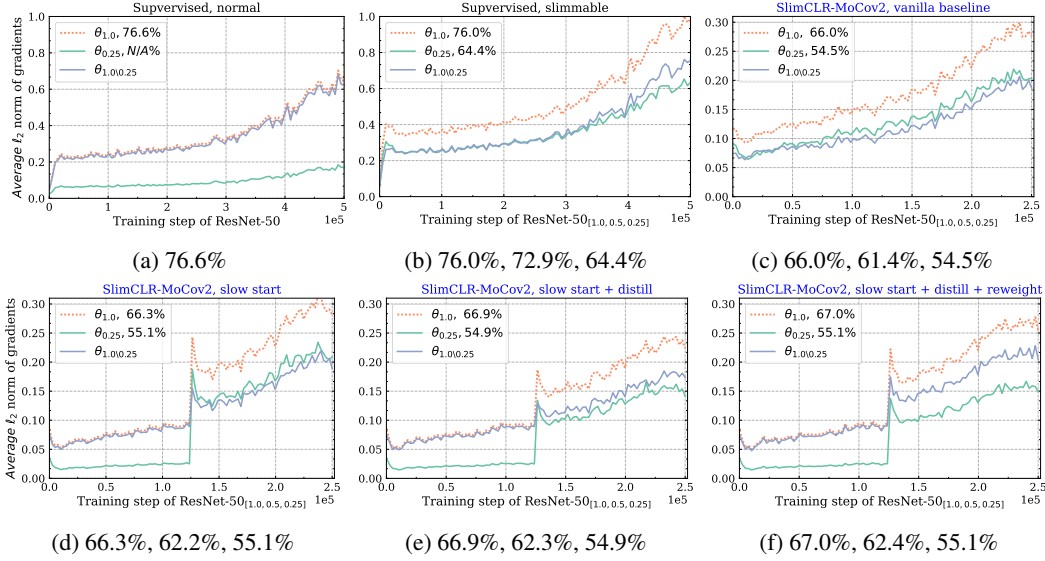

Figure 4: The average gradient norm. We calculate the $\ell_2$ norm of gradients layer-wise and get their mean. $\theta_{1.0} \in \Theta_{1.0}$ and $\theta_{0.25} \in \Theta_{0.25}$ represent parameters of the full model and sub-network with width 0.25. $\theta_{1.0\backslash 0.25} \in \Theta_{1.0}\backslash\Theta_{0.25}$, denote the rest parameters of $\Theta_{1.0}$ besides $\Theta_{0.25}$. It is normal that the gradient norm increases during training (Goodfellow et al., 2016).

where $C$ is the number of classes. Following classical KD (Hinton et al., 2015), we further assume $\sum_j z_j = \sum_j v_j = 0$, we can get:

$$\frac{\partial \mathcal{L}}{\partial z_i} \approx \frac{1}{C\tau_s} \left( \frac{z_i}{\tau_s} - \frac{v_i}{\tau_t} \right). \tag{33}$$

## A.3 VISUALIZATION

In this section, we provide more detailed visualization and explanations for gradient imbalance, gradient direction divergence, and optimization trajectory during training. These visualization can help readers better understand how our methods work. Plus, they are also helpful for understanding the difference of training slimmable networks in supervised and self-supervised learning.

### A.3.1 GRADIENT IMBALANCE

Besides the ratios of gradient norms in Figure 3, we also display the absolute values of gradient norms in Figure 4 to help readers better understand the gradient imbalance phenomenon.

### A.3.2 GRADIENT DIRECTION DIVERGENCE

Besides the imbalance of gradient magnitudes, the gradient directions of different weight-sharing networks also have conflicts with each other. Such conflicts result in disordered gradient directions of the full model, and we call the phenomenon gradient direction divergence in this work.

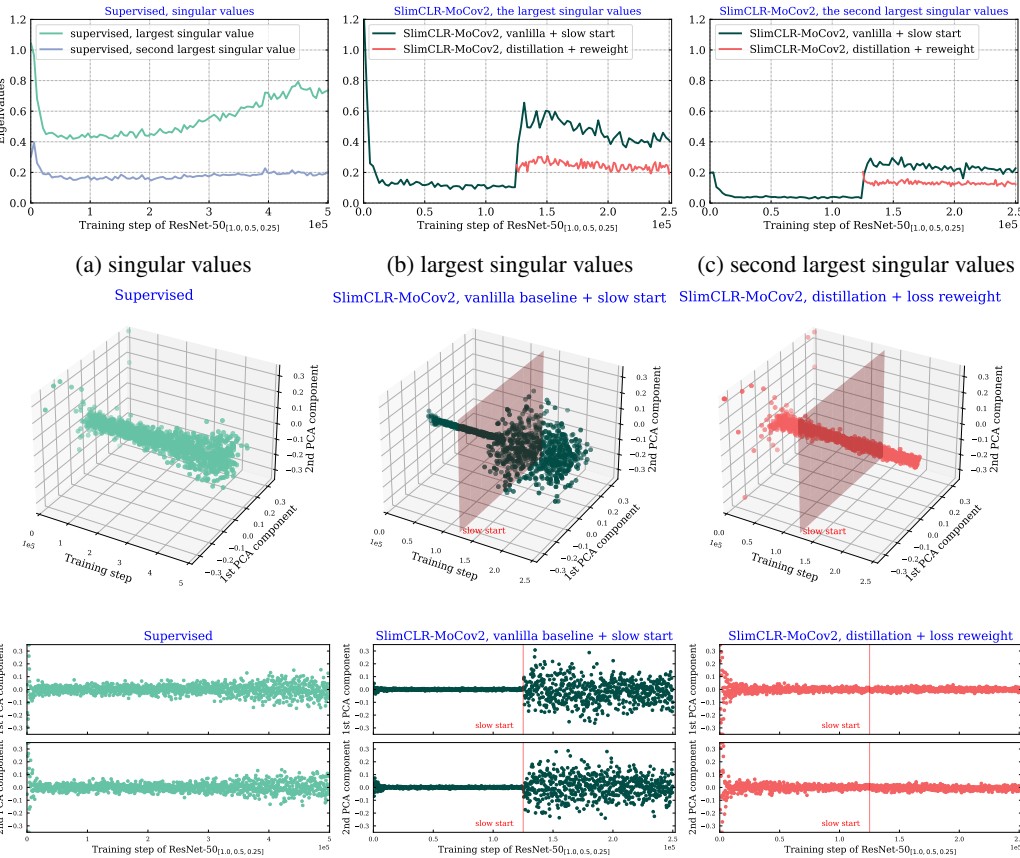

(a) singular values  (b) largest singular values  (c) second largest singular values

(d) Principal directions of gradients (e) Principal directions of gradients (f) Principal directions of gradients

Figure 5: The gradient direction divergence. (a) (b) (c) The singular values of the full gradient matrix of the last linear layer. (c) (d) (e) The principal directions of gradients of the last linear layer.

We show the phenomenon when training slimmable networks on ImageNet in Figure 5, supervised and self-supervised. In Figure 5a&5b&5c, the largest and second largest singular values of the full gradient matrix of the last linear layer in the model are displayed. In Figure 5b, the network has small eigenvalues without slimmable training, namely, no interference from other weight-sharing networks. In Figure 5b&5c, after the slow start point, we can see that training of weight-sharing networks dramatically increase the singular values. In supervised case (Figure 5a), training of weight-sharing networks also results in large singular values.

**Training of weight-sharing networks will make the directions of full gradients disordered, and such a phenomenon is more serious in self-supervised learning.** We visualize the gradient directions in Figure 5d&5e&5f. Specifically, we collect the gradients of weights of the last linear layer during training; after training, we perform PCA on these gradients and show their projections on the first two principal components (Li et al., 2018). In Figure 5e, we can see that the gradient directions are stable and consistent during training before slow start. However, after slow start, gradient directions become disordered due to conflicts of weight-sharing networks during training. In Figure 5d, training of weight-sharing networks also makes the gradient directions disordered. However, as the networks have the same global supervision, the gradient direction divergence is less obvious compared to Figure 5e. In Figure 5f, our proposed distillation and loss reweighting techniques effectively solve the divergence of gradient directions and make the training process stable.

### A.3.3 ERROR SURFACE AND OPTIMIZATION TRAJECTORY

The performance degradation caused by the interference between weight-sharing networks is more severe in self-supervised learning compared to supervised learning. This is because the gradient imbalance and gradient direction divergence are both more significant in self-supervised cases. In this subsection, we further provide the the error surface and optimization trajectory (Li et al., 2018) when training a slimmable networks to re-emphasize the point: **the optimization process of a slimmable network is harder in self-supervised cases than that in supervised cases**.

We train slimmable networks in both supervised and self-supervised (MoCo (He et al., 2020)) manners on CIFAR-10 (Krizhevsky & Hinton, 2009). The base network is a ResNet-20×4, which has 4.3M parameters. We train the model for 100 epochs. At the end of each epoch, we save the weights of the full model and calculate the Top-1 accuracy. For self-supervised cases, we use a $k$-NN predictor (Wu et al., 2018) to obtain the accuracy. After training, we calculate the principle components of the differences of the saved weights at each epoch and the weights of the final model following Li et al. (2018). Then we use the first two principle components as the directions to plot the error surface and optimization trajectory in Figure 6.

The visualization shows that self-supervised learning is harder than supervised learning. In the left error surface in Figure 6a and Figure 6c, we can see that the terrain around the valley is flat in supervised cases; by contrast, the terrain around the valley is more complicated in self-supervised cases. From the trajectory of ResNet-20×4 in the left of Figure 6b and Figure 6d, the contours in supervised cases are denser, *i.e.*, the nearby two contours are closer. Namely, the model in self-supervised cases costs more time to achieve the same improvement of accuracy compared to the model in supervised cases (the gaps between two contour lines are all the same). In supervised cases, clear global guidance helps the model quickly reach the global minima. In self-supervised cases, it is harder for the model to reach the global minima fast without such global guidance.

The visualization shows that the interference of weight-sharing networks is more significant in self-supervised cases. First of all, in self-supervised cases, weight-sharing networks bring huge changes to the error surface in Figure 6c. In contrast, the change is not so obvious in supervised cases. Second, the interference between weight-sharing networks in self-supervised cases makes the model shift more away from the global minima (the origin in the visualization) as shown in Figure 6d. In Figure 6b, the maximal offsets from the global minima along the 2nd PCA component are 21.75 and 28.49 for ResNet-20×4 and ResNet-20×4$_{[1.0,0.5]}$. The offset increased 31.0%. For self-supervised cases in Figure 6d, the maximal offsets from the global minima along the 2nd PCA component are 13.26 and 18.75 for ResNet-20×4 and ResNet-20×4$_{[1.0,0.5]}$. The offset increased 41.4%. It is clear that the interference of weight-sharing networks is more significant in self-supervised cases compared to supervised cases.

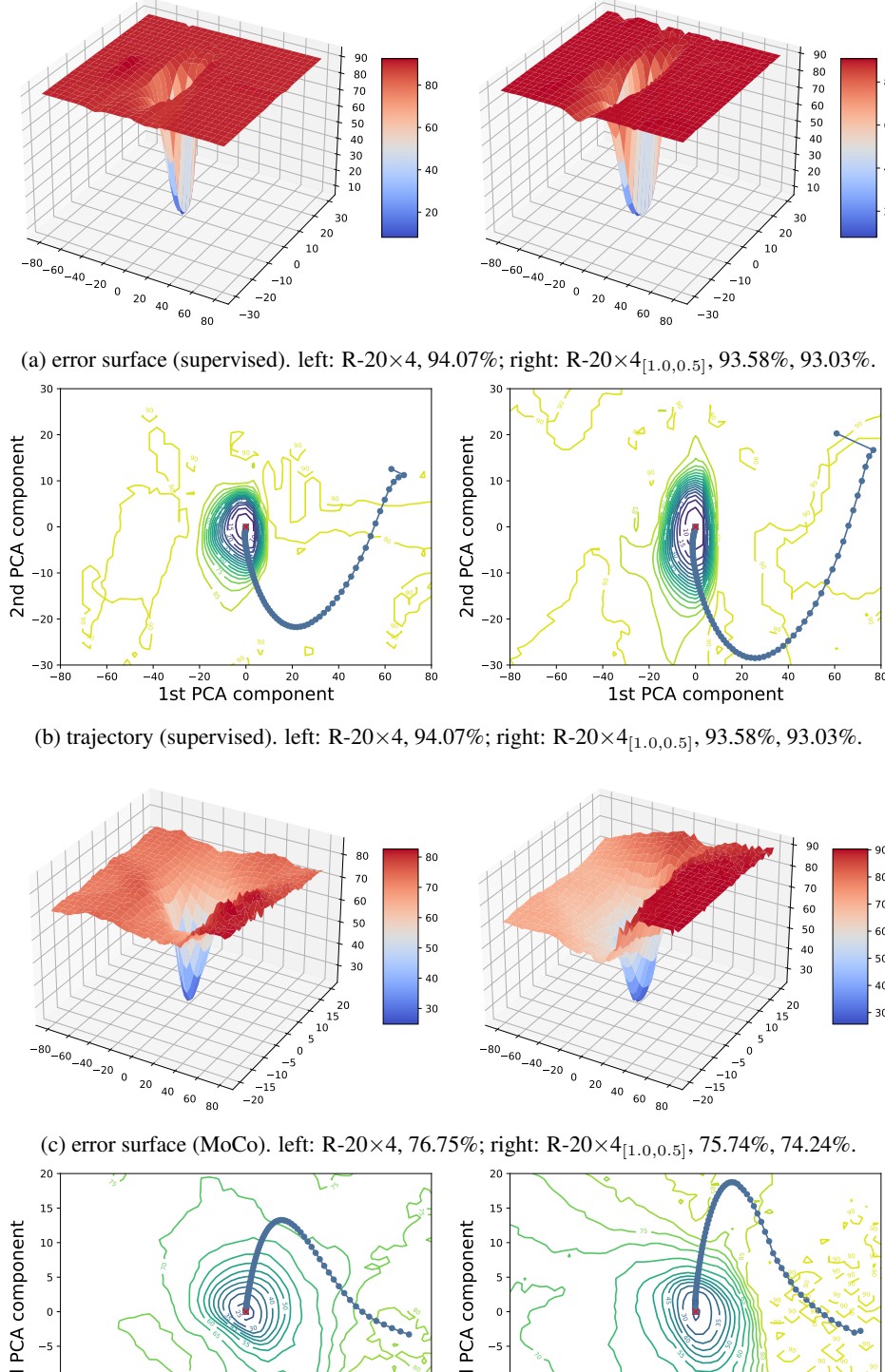

(a) error surface (supervised). left: R-20×4, 94.07%; right: R-20×4$_{[1.0,0.5]}$, 93.58%, 93.03%.

(b) trajectory (supervised). left: R-20×4, 94.07%; right: R-20×4$_{[1.0,0.5]}$, 93.58%, 93.03%.

(c) error surface (MoCo). left: R-20×4, 76.75%; right: R-20×4$_{[1.0,0.5]}$, 75.74%, 74.24%.

(d) trajectory (MoCo). left: R-20×4, 76.75%; right: R-20×4$_{[1.0,0.5]}$, 75.74%, 74.24%.

Figure 6: Visualization of the error surface and optimization trajectory.

| Model | $\tau_2 = 5.0$ | | $\tau_2 = 1.0$ | |
|---|---|---|---|---|
| | Top-1 | Top-5 | Top-1 | Top-5 |
| R-50$_{1.0}$ | 69.0 | 89.0 | **69.2** | **89.1** |
| R-50$_{0.75}$ | 64.0 | 86.0 | **64.5** | **86.3** |
| R-50$_{0.5}$ | 62.5 | 84.8 | **62.8** | **85.0** |
| R-50$_{0.25}$ | 57.2 | 80.8 | **57.2** | **80.9** |

(a) temperature $\tau_2$, 100 epochs

| Model | 1.2 | 2.8 | 3.2 | 3.6 |
|---|---|---|---|---|
| | | Top-1 | | |
| R-50$_{1.0}$ | 70.7 | 71.9 | **72.3** | 72.1 |
| R-50$_{0.75}$ | 66.8 | 69.4 | 69.6 | **69.8** |
| R-50$_{0.5}$ | 64.7 | 67.2 | 67.6 | **68.0** |
| R-50$_{0.25}$ | 59.1 | 61.8 | 62.1 | **62.4** |

(b) $lr$ for slow start, 300 epochs

Table 4: Ablation experiments with SlimCLR-MoCov3 on ImageNet.

## A.4 MORE IMPLEMENTATION DETAILS

**Slimmable networks** We adopt the implementation of slimmable networks described in Yu et al. (2019), which has switchable batch normalization layers. Namely, each network in the slimmable network has its own independent batch normalization process.

**SlimCLR-MoCov2** We train SlimCLR-MoCov2 on 8 Tesla V100 32GB GPUs without synchronized batch normalization across GPUs. The momentum coefficient $m$ is 0.999 during training.

**SlimCLR-MoCov3** We train SlimCLR-MoCov3 on 8 Tesla V100 32GB GPUs with synchronized batch normalization across GPUs. Synchronized batch normalization is important for MoCov3 to obtain a better performance in linear evaluation. The momentum coefficient $m$ is 0.99 with a cosine schedule when training for 300 epochs. The data augmentations are the same as augmentations of MoCov3 Chen et al. (2021).

**Ablation study with SlimCLR-MoCov3** Besides the ablation studies with SlimCLR-MoCov2, we also provide empirical analysis for SlimCLR-MoCov3. Different from SlimCLR-MoCov2, SlimCLR-MoCov3 adopts different temperatures $\tau_1 = 1.0$ and $\tau_2 = 1.0$. When applying slow start, SlimCLR-MoCov3 also increases the initial learning rate at the same time.

In Table 4a, we show the influence of temperature $\tau_2$ for online distillation. Different from the choice in SlimCLR (MoCov2), it is better for SlimCLR-MoCov3 to choose a temperature $\tau_2$ for online distillation, which is close to the temperature $\tau_1$ of contrastive loss. $\tau_1 = 1.0$ is the default choice of MoCov3 (Chen et al., 2021), and we do not modify it.

Another interesting phenomenon is that when the number of training epochs of SlimCLR-MoCov3 becomes larger, we need to increase the learning rate ($lr$) when we start to train the sub-networks. The influence of learning rate for slimmable training in SlimCLR-MoCov3 is shown in Table 4b. Here the learning rate refers to the base learning rate, the immediate learning rate after the warm-up is calculated by this base learning rate and the current training steps: $\frac{1}{2}lr(1 + \cos(\pi \frac{step}{max\_step}))$. Different from SlimCLR-MoCov2 and SlimCLR-MoCov3 with fewer epochs, SlimCLR-MoCov3 will get poor performance if we do not change the learning rate when training for more epochs. We attribute the difference to the LARS (You et al., 2017) optimizer we adopt for SlimCLR-MoCov3. LARS normalizes the gradient of layers in the networks to avoid the imbalance of gradient magnitude across layers and ensures convergence when training networks with very large batch size. LARS is sensitive to the change in learning rate and helps self-supervised models training with large batches converge fast (Chen et al., 2021; 2020a). When training for the 300 epochs, the full model can reach a local minima fast in the first 150 epochs. In this case, a learning rate 0.6 (half of the base learning rate) is not able to help the system walk out of the valley and reach a better local minima. Consequently, a large learning rate is needed to give the system more powerful momentum. From Table 4b, we can also see that SlimCLR-MoCov3 with LARS is sensitive to the change in learning rate. This is consistent with observations of previous works (Chen et al., 2021; 2020a).

