# OpenReview forum: "Slimmable Networks for Contrastive Self-supervised Learning"
_ICLR.cc/2023/Conference — Submitted to ICLR 2023_

### Official Review · Reviewer_5mjb · 2022-10-21

**Confidence:** 3
**Correctness:** 3
**Technical Novelty And Significance:** 1
**Empirical Novelty And Significance:** 2
**Recommendation:** 3

**Clarity, Quality, Novelty And Reproducibility:**

The detailed comments are described in the previous section. In summary,
- Clarity :: This proposed idea is clear, but its description is hard to follow in general.
- Quality :: The empirical results are somewhat strong, but it is not clear where the performance gains come from (proposed techniques or MoCo-v3).
- Novelty :: This paper lacks methodological novelty.
- Reproducibility :: This paper describes the implementation details well.


**Strength And Weaknesses:**

Strengths
- I think the main strength of this paper is that the proposed method outperforms other distillation approaches.

Weaknesses
- I feel the lack of methodological novelty. The idea of training slimmable neural networks is not new, and online distillation was already used in [1]. I also think other techniques are just engineering, and they have often been utilized in other literature: e.g., loss reweighting is a common strategy for multi-task learning.
- Although this paper obtains some gains from the proposed techniques, the gains are marginal, and most of the gains come from using a large number of training epochs or MoCo-v3, not the techniques.
- The training cost is linearly increasing with respect to the number of sub-networks. This could be critical since self-supervised learning is often time-consuming.
- The motivation, the gradient imbalance, is really critical? I think the motivation is somewhat weak: the gradient imbalance can occur in any neural network layer because there exist dominant and minor neurons in any layer (e.g., think about singular value decomposition of weight matrix). Training sub-networks can be considered as determining the order of neuron importance. Note that the result in Fig (3a) is due to the random order of the neurons.
  - Could you provide the distribution of singular values of the full weight matrix instead of gradient norms? I think it would be better to understand how training slimmable networks affect the weight matrix.
- This paper is hard to follow: there are many confusing notations and descriptions.
  - It would be better to use colors in Figure 1. In addition, it would be better to include the results of the proposed method in Figure 1 for readers.
  - In general, notations should be defined before using them. For example, there is no description of ξ when using the parameter.
  - Suggest to use $\Theta_{w_i}=\\{\theta_{w_1},\ldots,\theta_{w_i\}\\}$ instead of writing $\Theta_{w_j}\subset\Theta_{w_i}$ if $w_j<w_i$.
  - In Eq (1), why $\xi_1$ instead of $\xi_i$? There is no explanation for this.
  - In the self-supervised learning literature, the first MLP is often referred to as projection and the second MLP as prediction. I recommend following the conventional terminologies. For example, SlimCLR-MoCo-v2 should use projection instead of prediction. The current usage causes some confusion.
  - What is $\theta_{1.0\setminus0.25}$?
  - What are the main and minor parameters? And why are they main and minor?

[1] Yu & Huang, Universally Slimmable Networks and Improved Training Techniques, 2019


**Summary Of The Paper:**

This paper aims to learn slimmable neural networks with contrastive self-supervised learning without labels. To further improve the performance of sub-networks, this paper suggests using (i) slow-start training for sub-networks, (ii) online distillation, and (iii) loss reweighting. This increases the gradient norm of the main parameters, leading to performance improvements in the main network. This paper demonstrates that the main and sub-networks outperform existing two-stage distillation approaches.


**Summary Of The Review:**

While learning slimmable neural networks is interesting, I feel the lack of methodological novelty and the weak motivation about gradient imbalance. In addition, this paper is hard to follow with confused notations. Hence I vote for rejection.

---

> ### Author Response · Authors · 2022-11-18
> **response (1/2)**
>
> **1. lack of methodological novelty.**
>
> Please do not ignore the main contribution of the paper: introducing a new solution (slimmable networks) to get self-supervised small models, finding the problems during this process, and making it work perfectly in self-supervised learning. It is not so “simple”. It is new to introduce slimmable networks into self-supervised learning, and we are the first ones to do such a thing. It is also non-trivial to find the gradient imbalance and gradient direction divergence phenomenon of training slimmable networks in self-supervised learning (refer to Appendix A.3 VISUALIZATION). Borrowing the wisdom of previous works to solve new problems is not something wrong. We believe our work brings new findings to the community and may inspire some following works.
>
>
> Besides, we show great respect for the previous works and already pointed out this point in introduction and Sec.3.2., online distillation is inspired by US-Nets. One difference is that inplace distillation uses the features of networks to do distillations. In our papers, we use the similarity distributions of negative and positive samples to do distillations. This is better than using the feature in self-supervised learning, experiments are available upon request. Please check Eq.(4) in the paper and related descriptions.
>
> **2. the gains are marginal, and most of the gains come from using a large number of training epochs or MoCo-v3, not the techniques**
>
> The “marginal” claim is somehow contradictory to the strengths “outperforms other distillation approaches”.
>
> - Please check Table 2 to see how much we improve the baseline. Without such efforts, MoCov3 will still suffer from serious performance degradation.
>
> - Please do not ignore the effort of previous works. Check Compress/SEED/DisCo/BINGO (in Table 1) to see what they do for better performance. They cannot get published by simply combining MoCov2 and knowledge distillation.
>
> - We do not use longer training epochs. In Table 1, for SlimCLR-MoCov3, the slow start epoch of sub-networks is 150, actually, the small networks in SlimCLR-MoCov3 are only trained for 150 epochs. Besides, considering BINGO R-18 with R-152 as the teacher, the training epoch of R-152 is larger than 800 and the training epoch of R-18 is 200. Please re-consider the total training time and cost.
>
> **3. The training cost is linearly increasing with respect to the number of sub-networks**
>
> The training time is not linearly increasing w.r.t. the number of sub-networks. Sub-networks are smaller than the full network and cost less time to train. Slow start also saves much time during training. Specifically (provided in Sec 4.3), the pre-training time of SlimCLR-MoCov2 without and with slow start epoch S = 100 on 8 Tesla V100 GPUs are 45 and 33 hours, respectively. For reference, the pre-training time of MoCov2 with ResNet-50 is roughly 20 hours. We only cost ~50\% additional training time but get 4 different networks that are suitable for different devices.
>
> **4. The motivation, the gradient imbalance, is really critical?**
>
> As we showed in Figure 3 and Table 2, gradient imbalance is really critical.
>
> What is the meaning of "the random order of the neurons"? Please check the caption of Figure 3 to find out how we calculate the ratios of gradient norms. No random sampling process during the calculation. We calculate the gradient norm of each layer and then average these gradient norms layer-wise, and the final average result will be used to calculate the ratios. We believe such an average gradient norm is a persuasive metric.
> As for the dominant and minor neurons, please provide some related literature and we can better give a response. Besides, we believe dominant and minor neurons do not matter considering our carefully designed calculation manner of gradient norms.
>
> - distribution of singular values of the full weight matrix
>
> Please check the newly added A.3.2 GRADIENT DIRECTION DIVERGENCE. Besides the singular values, we also provide visualizations of gradient directions.

---

> > ### Author Response · Authors · 2022-11-18
> > **response (2/2)**
> >
> >
> > **5. This paper is hard to follow**
> >
> > Some confusion may be caused by a quick look-through.
> >
> > - Please check the caption of Figure 2: “$\xi$ are an exponential moving average of $\theta$.”.
> >
> >
> > - Please check the start paragraph of Sec 3.1 DESCRIPTION OF SLIMCLR. As the reviewer suggests, $\Theta_{w_i}=\{\theta_{w_1},\ldots,\theta_{w_i}\}$, first, as we said in the paper, $w_1$ is the largest width (the width of the full model), so $\theta_{w_1}$ cannot belong to $\Theta_{w_i}$ with $w_i < w_1$; second, if we use $\Theta_{w_i}=\{\theta_{w_i},\ldots,\theta_{w_n}\} $, this indicates an exclusive relationship of $\theta_{w_i}$ and $\theta_{w_n}$. When we say a parameter belongs to  $\Theta_{w_i}$, we need to pay more attention to tell the readers whether this parameter is $\theta_{w_i}$ or $\theta_{w_n}$ or any other parameters. This is tedious and unnecessary. We believe the now-used notation is a better choice.
> >
> > - Please check the description above Eq.(1) in the paper: “for the second view…”. It is pretty straightforward to use only the largest model for the momentum branch. Otherwise,  additional training time and cost will be introduced. Without the same momentum encoder, training will also become harder.
> >
> >
> > - This slightly different choice for the simplicity of math notations. Imagine using projection as the name and changing the math notations for SlimCLR-MoCo-v2, more math equations are needed to clearly define loss of  SlimCLR-MoCo-v3. Given Figure 2, we believe it is enough to understand the whole process of SlimCLR-MoCo-v2 and SlimCLR-MoCo-v3.
> >
> > - Please check the caption of Figure 3 or the second paragraph of 3.2 GRADIENT IMBALANCE AND SOLUTIONS.
> > "parameters $\theta_{1.0 \backslash 0.25} \in \Theta_{1.0} \backslash \Theta_{0.25}$, \ie, rest parameters of $\Theta_{1.0}$ besides $\Theta_{0.25}$"
> >
> > - Please check the second paragraph of 3.2 GRADIENT IMBALANCE AND SOLUTIONS.
> > Generally, main parameters mean a large proportion of parameters, at least more than 50\% of the parameters. In the paragraph and Figure 3, the main parameters mean more than 90\% of the parameters, and the minor parameters mean less than 10\% of the parameters. We give the ratios of the numbers of parameters in the second paragraph of 3.2.

---

### Official Review · Reviewer_1XPr · 2022-10-23

**Confidence:** 5
**Correctness:** 3
**Technical Novelty And Significance:** 1
**Empirical Novelty And Significance:** 2
**Recommendation:** 3

**Clarity, Quality, Novelty And Reproducibility:**

From the technical part, the paper lacks novelty and I don’t see techniques customized to train slimmable networks in the self-supervised setting. From the experimental perspective, it lacks essential empirical studies. Overall, the paper is clearly below the acceptance threshold.

**Strength And Weaknesses:**

### Strength

This paper is a good study of the effect of slimmable networks for contrastive self-supervised learning. The empirical analysis has some contribution to the community.

### Weakness

**1: The technical novelty is limited and some details are confusing.**

Slimmable networks are a special case of widely studied one-shot NAS (e.g., [R2, R3, R4, R5, R6]), which only considers the width dimension (see discussion in OFA [R2]). There are many techniques to deal with interference among subnetworks. Specifically,
* “Slow start” belongs to progressive training in one-shot NAS. For example, OFA proposes a “Progressive Shrinking” strategy, which starts with training the largest sub-network and then progressively fine-tunes the network to support smaller sub-networks by gradually adding them into the sampling space.

 * “Online distillation” was originally proposed in US-Nets [Yu et al., 2019b]. Apart from the inplace distillation, it also proposes the sandwich rule.

* “loss reweights” aims to assign larger weights for sub-networks with large widths. However, it violates the training objective of slimmable networks. The objective is to make each supported sub-network maintain the same level of accuracy as independently training a network with the same architectural configuration, rather than only training an accurate large “supernet”. This is evidenced in Table 2 (e), where adding loss reweighting makes R-50(0.25) perform worse, so what’s the meaning there?

**2:  Another concern is what are the fundamental differences between self-supervised and supervised training for slimmable networks?** This is not clear to me as all the training techniques used are common practices in supervised training.

**3: What’s the relationship between unsupervised NAS (e.g., [R4, R5]), including the contrastive self-supervised one (e.g., [R6])?**

4: In Page 8, the authors study 4 possible cases of loss reweighting and show the results in Table 2e. However, I find case (3) archives the best performance for most widths but the paper uses case (1) by default in Eq. (5). I disagree with the author's explanation that “To ensure the performance of the smallest network, we adopt the reweighting manner (1) in practice” as all sub-networks with different widths should be equally important. Otherwise, what’s the meaning of slimmable networks there?

**5: The paper lacks mathematical modeling for the gradient divergence issue which leads to the optimization difficulty claimed by the authors.** I think there are only four possible widths and it is not difficult to analyze the gradient magnitude and directions using SGD with maths formulations. Also, some theoretical analysis on convergence is expected, even assuming a linear neural network is fine [R1].

**6: The experiments are far from enough to justify the effectiveness of the proposed method.**

* 6.1:  The results are merely based on the ResNet-50 backbone. However, I would like to see more ResNet backbones such as R-101 and R-152. More importantly, experiments on Vision Transformers, such as ViT-B in MoCo v3, must be included in the experiments.

* 6.2:  The paper only evaluates the representation quality using linear probing. However, it must evaluate transfer learning performance which is the standard practice in self-supervised learning (e.g., in MoCo v3). For example, experiments on dataset transfer and downstream tasks such as dense segmentation and detection on COCO and ADE20k are needed.

* 6.3: How about training the whole network (width 1.0) first then using network pruning (e.g., [R7]) to obtain small networks (width 0.25, 0.5, 0.75)? As this strategy can avoid the interference issue during training.

* 6.4: It lacks comparisons with methods dealing with sub-network interference, such as switchable BN [Yu et al., 2019], sandwich rules [Yu et al., 2019b] and many others.

**7:  The discussions and references in related work are far from enough.** There are few discussions with single-shot NAS and unsupervised NAS methods. In addition, as I point out in the technical novelty part, the differences and advantages with the related work must be discussed.

**8:  Writing also needs to be improved.**

* 8.1: What is the definition of the “main parameters” in the introduction?

* 8.2: In Sec. 3.2, “..., where $L$ is the loss function”. It should be defined in Eq. (1) where it first appears.

* 8.3: Many grammar issues. I only point out a few. “Slimmable neworks” in Sec. 2; “server performance degradation” in Sec. 3.2.

9: In Sec. 3.2, authors argue that the two ratios in Figure 3 should be large enough. “In Figure 3f, ..., are larger than 1.0 by a clear margin”. It does provide a clear concept of how large is good enough. In my opinion, it also depends on the network architectures and self-supervised learning frameworks. So Figure 3 may not be statistically significant.


**References:**

[R1]: “On the optimization of Deep Networks: Implicit Acceleration by Overparameterization”, ICML 2018

[R2]: “ONCE FOR ALL: TRAIN ONE NETWORK AND SPECIALIZE IT FOR EFFICIENT DEPLOYMENT”, ICLR 2020

[R3]: “BigNAS: Scaling Up Neural Architecture Search with Big Single-Stage Models”, ECCV 2020

[R4]: “Are Labels Necessary for Neural Architecture Search?”, ECCV 2020

[R5]: “Does Unsupervised Architecture Representation Learning Help Neural Architecture Search?”, NeurIPS 2020

[R6]: “Contrastive Self-supervised Neural Architecture Search”, Arxiv 2021

[R7]: “Resrep: Lossless cnn pruning via decoupling remembering and forgetting”, CVPR 2022

**Summary Of The Paper:**

This paper studies slimmable networks under the contrastive self-supervised learning (SlimCLR) setting. It proposes some strategies to solve the interference between weight-sharing sub-networks during training, including slow start training of sub-networks, online distillation, and loss re-weighting according to model sizes, and a switchable linear probe layer. SlimCLR is evaluated on ImageNet based on ResNet-50 using MoCo v2 and v3 frameworks.

**Summary Of The Review:**

Please refer to the above.

---

> ### Author Response · Authors · 2022-11-18
> **response (1/3)**
>
> **0. before the point-to-point response**
>
> First of all, we would like to thank the reviewer for the detailed comments. However, we believe the reviewer messed up two different concepts: architecture learning and representation learning. This leads to an underestimation of the contribution of this paper. Besides, the reviewer makes a strong assumption that supervised methods (in NAS) will easily transfer to self-supervised learning and work normally without labels. This assumption is too strong and does not hold in most cases, i.e., slimmable networks will get severe performance degradation in self-supervised learning.
> Next, we will give a point-to-point response and make further explanations about what we say above.
>
> **1. The technical novelty is limited and some details are confusing.**
>
> After checking [R2, R3, R4, R5, R6], we find only [R2, R3] belong to one-shot NAS, [R4, R5, R6] are not one-shot NAS. In our perspective, [R4, R5, R6] are normal NAS methods equipped with a self-supervised pretext during the searching stage, and they also needed to be evaluated under supervision (train in a normal way like training a ResNet in supervised learning).
> A point needed to be emphasized: what [R2, R3, R4, R5, R6] do is architecture learning, they try to find the best architecture in certain cases. By contrast, what we do is representation learning, we try to learn better representations given certain architectures. R5 may do representation learning and architecture learning at the same time. They focus on exploring the relationships between learned representations and the performance of the architecture on other downstream tasks. Nevertheless, their settings and tasks are way different from the main-stream methods and our paper.
>
> **1.1 “Slow start” belongs to progressive training in one-shot NAS.**
>
> Progressive shrinking and BigNAS are missing related works and we add them to the revision. However, this does not mean OFA[R2] or BigNAS[R3] will work in a self-supervised case. In Figure.1 in the paper, we can see that the performance degradation will become more severe when the number of weight-sharing networks increases. Slimmable networks also behave differently in supervised learning and self-supervised learning as shown in Figures 3b\&3c and Appendix A.3 VISUALIZATION.
>
> **1.2 “Online distillation” was originally proposed in US-Nets**
>
> We show great respect for the previous works and already pointed out this point in introduction and Sec.3.2., online distillation is inspired by US-Nets. One difference is that inplace distillation uses the features of networks to do distillations. In our papers, we use the similarity distributions of negative and positive samples to do distillations. This is better than using the feature in self-supervised learning, experiments are available upon request. Please check Eq.(4) in the paper and related descriptions.
>
> **1.3 what’s the meaning of “loss reweights”**
>
> This is a misunderstanding. Please check Table 2d\&2e, experiments in Table 2d with temperature 5.0 are baselines for experiments in Table 2e. For case (1), R-50(0.25) achieves 55.1\% accuracy, which is better than the baseline 54.9\%.
>
> Loss reweighting is not in conflict with the goal of slimmable networks. Loss reweighting is helpful to solve the networks interference and help all weight-sharing networks achieve better performance. Please check Figure Table 2d\&2e for a comparison. The visualization of the effect of loss reweighting can be found in Figure 3 and Figure 5.
>
> **1.4.**
>
> At the end of this point, we want to emphasize that this paper is a pioneer work to introduce slimmable networks into self-supervised learning. We provide a new solution to get self-supervised small models with good performance without other teachers. We also provide a detailed analysis of why slimmable networks do not work normally in self-supervised learning. We believe our work makes meaningful contributions to the community and may inspire many following works.
> We are not shamed for borrowing wisdom from previous literature. We appreciate their works, but our contributions are also unique.

---

> > ### Author Response · Authors · 2022-11-18
> > **response (2/3)**
> >
> > **2. differences between self-supervised and supervised training for slimmable networks**
> >
> > This is a good question and we add a new paragraph to discuss it before the Sec. CONCLUSION in the revision.
> >
> > We also provide some answers to this question to some extent.
> > In Appendix A.3, through extensive visualizations, we show that optimization process of training
> > slimmable networks is harder in self-supervised cases than that in supervised cases. Specifically, the
> > gradient imbalance and gradient direction divergence are both more significant in self-supervised
> > cases as we discussed in Section 3.2 and Appendix A.3.2. In supervised cases, clear global supervision
> > can help slimmable networks avoid these problems to some extent during training. In
> > self-supervised learning, we do not have such clear global supervision, and we need to pay more
> > effort to deal with the performance degradation problems.
> > In Appendix A.3.3, we further provide the error surface and optimization trajectory when training slimmable networks to help readers understand our points.
> >
> > To fully figure out this problem theoretically or empirically, we believe we first need to figure out what is the fundamental differences between self-supervised and supervised networks. As far as we know, nobody answers this question. We also hope this paper can raise the interest of people in this topic.
> >
> > **3. What’s the relationship between unsupervised NAS**
> >
> > [R4, R5], including [R6], they merely use a self-supervised pretext to search architectures, during evaluation, they still need to train from scratch with labels. They just borrow the wisdom of self-supervised learning to search architecture. They are doing architecture learning and have few relationships with our paper.
> >
> > **4. 4 possible cases of loss reweighting why case (1)**
> >
> > The baseline of Table 2e is Table 2d. When using case (1), most weight-sharing networks gain improvement. Besides, if we just compare case (1) and case (3) in Table 2e,
> > (67.4 - 67.5) + (65.5 - 65.9) + (62.5 - 62.6) + (55.1 - 54.4) = 0.1. Actually, the difference is trivial (case (1) is slightly better), especially considering they both get improvements compared to the baseline in Table 2d.
> >
> > **5. The paper lacks mathematical modeling**
> >
> > Actually, We provide theoretical results to explain the working mechanism of slimmable models in linear
> > cases. We also use such results to explain why a single slimmable proble layer does not work in practice.
> > Please check Appendix A.1.
> > We also check [R1], it demonstrates overparameterization of linear neural networks can accelerate convergence due to the increased eigenvalues (claims 1). I think only their formulation of problems can be learned, and it is similar to what we do in Appendix A.1.
> >
> > Also, please check A.3 VISUALIZATION, we provide extensive visualizations to help readers understand the gradient imbalance, gradient direction divergence, and optimization trajectory when training slimmable networks.
> >
> > Besides, we think the reviewer is a little harsh on this point. Given the 7 recommended papers, we have not seen one achieves good empirical results as well as beautiful theoretical results. We believe our provided analysis is enough for people to understand how to train a slimmable network in practice. These empirical or theoretical analyses are also new in this area.
> > It is also not so easy to say “I think … it is not difficult…”. A classical example like: “We propose that a 2 month, 10 man study of artificial intelligence be carried out during the summer of 1956 ……It is not difficult to design a machine which exhibits the following type of learning…..” — M. L. Minsky [x1]
> >
> > [x1] A PROPOSAL FOR THE DARTMOUTH SUMMER RESEARCH PROJECT ON ARTIFICIAL INTELLIGENCE http://jmc.stanford.edu/articles/dartmouth/dartmouth.pdf

---

> > > ### Author Response · Authors · 2022-11-18
> > > **response (3/3)**
> > >
> > > **6. The experiments are far from enough to justify the effectiveness of the proposed method.**
> > >
> > > **6.1 The results are merely based on the ResNet-50 backbone.**
> > >
> > > Recall the motivation of this paper and what we are trying to do: get good self-supervised pre-trained small models. With a ResNet-50 as the full network, we surpass previous distillation methods. It is meaningless to use a large full model like R-101 and R-152. As for ViT, we plan to add it in a journal version (if possible).
> > >
> > > **6.2. The paper only evaluates the representation quality using linear probing.**
> > >
> > > Please check the newly added Table 3 in the revision. We add transfer learning results for object detection and instance segmentation tasks. The pre-trained model of SlimCLR outperforms the supervised baseline in these two tasks on COCO 2017.
> > >
> > > **6.3 network pruning**
> > >
> > > We believe this is also a possible solution, but pruning will also introduce additional pruning cost. Besides, it does not conflict with our method.
> > >
> > > **6.4 It lacks comparisons with methods dealing with sub-network interference**
> > >
> > > Please check Appendix A.4. We already use switchable BN. Besides, we think sandwich rules do not aim for dealing with sub-network interference. It aims to guarantee the upper bound and lower bound of networks and avoid training all sub-networks. When the number of networks increases, interference becomes more serious as we said. Sandwich rules may not work in such cases.
> > >
> > > **7. The discussions and references in related work are far from enough**
> > >
> > > Some techniques in one-shot NAS are missing, we add them to the revision. However, as aforementioned, unsupervised NAS has no relationship with this paper. They are doing architecture learning, while we are doing representation learning.
> > >
> > >
> > > **8. Writing also needs to be improved.**
> > >
> > > **8.1 definition of the “main parameters”**
> > >
> > > A large part of parameters, more than 50\% of the parameters.
> > > We add more definitions in Sec.3.2 GRADIENT IMBALANCE AND SOLUTIONS.
> > >
> > > **8.2 $\mathcal{L}$**
> > >
> > > $\mathcal{L}$ is the actual training loss. It is not equal to Eq.(1). $\mathcal{L}$ can also be Eq.(6) in Figure 3f, or other variants. This is a general reference to training loss.
> > >
> > > **8.3 typo**
> > >
> > > Thanks. We corrected them.
> > >
> > > **9.  it also depends on the network architectures and self-supervised learning frameworks**
> > >
> > > ResNet is a typical choice in self-supervised learning. Given Eq.(3) in the paper and Figure 3, we think it is enough to understand the gradient imbalance for the ResNet family. We are the first one to design a self-supervised learning framework for slimmable networks. We also hope the follower may have other new findings or overturn our conclusions. However, as a pioneer, we believe this is not a problem for this paper.

---

### Official Review · Reviewer_enLj · 2022-10-25

**Confidence:** 4
**Correctness:** 3
**Technical Novelty And Significance:** 2
**Empirical Novelty And Significance:** 3
**Recommendation:** 5

**Clarity, Quality, Novelty And Reproducibility:**

Clear, and well-written. But the novelty seems limited to me, with no theory, and the hypothesis seems not convincing. Reproducibility should be ok.

**Strength And Weaknesses:**

An interesting work and clear motivation.

However, I am not convinced about the gradient imbalance in learning slimmable networks. The measurement used in the paper seems not sufficient to me. Imagine that for each iteration the network weights are updated in a cyclic way, i.e., weights at (1,2,3) are updated in the first iteration, then (4,5,6), (7,8,9), (10,11,12) .... In this way the gradient norm ratio will be low as well, but the network can still be trained reasonably. Another question is since the paper mentioned Res18 in the motivation, is there a comparison of R-18 compared with R-18 distilled by R-50 teacher?

**Summary Of The Paper:**

Motivated by the performance gap between self-supervised learning and its counter supervised model, this work provides a one-stage self-supervised small model pretraining protocol. The slimmable network idea is used to get the representations of one augmentation $x_1$. Weights are shared between the set of slimmable encoders (the small one shared with the larger one). All the outputs from slimmable encoders are used as anchors in InfoNCE. The sum of loss from different anchors is calculated eventually. However, this naive implementation will cause gradient imbalance. The work proposed updating the full encoder first, knowledge distillation from the full encoder to the sub-network, and loss reweighting to mitigate the imbalance issue.

**Summary Of The Review:**

Overall this is an interesting work even though the idea of slammable is not new and no theory is provided. I hope the author could address my questions in the rebuttal. I’m wondering whether a similar range of performance drops will happen on R-18 like on R-50.

---

> ### Author Response · Authors · 2022-11-18
> **response**
>
> **1. The measurement used in the paper seems not sufficient**
>
> We are not sure we fully understand what the reviewer said. Considering ResNet-50 with widths [1.0, 0.75, 0.5, 0.25], if we update the four networks in a cyclic way, e.g., first R-50(1.0), then R-50(0.75), the gradient of R-50(0.75) will also break the updated state of R-50(1.0) because of the weight-sharing mechanism. Please check Eq.(3), the gradient imbalance is decided by the multiple losses and sharing weights. Once the weight-sharing networks have multiple losses and they share weights, some of the shared weights will definitely receive gradient from different losses and leading to gradient imbalance. This cannot be avoided by some training tricks.
>
> In the revision, we add one more measurement to observe the gradient directions in A.3.2 GRADIENT DIRECTION DIVERGENCE.
> It shows that the gradient directions of different weight-sharing networks have conflicts with each other.
> Such conflicts result in disordered gradient directions
> of the full model.
> Hope such supplementary can help the reviewer better understand how slimmable networks behave in self-supervised learning.
>
> **2.  is there a comparison of R-18 compared with R-18 distilled by R-50 teacher**
>
> Well, we do not have the results of R-18. However, in Table 1, we achieve much better results than R-18 with R-50 or R-152 as teachers. Specifically, our R-50(0.5) achieves 67.6\% top-1 accuracy on ImageNet during linear evaluation, which is significantly better than R-18 with R-50 or R-152 as teachers with fewer parameters and FLOPs.
>
> If the reviewer means using R-18 as the full model, it will also suffer from gradient imbalance and performance degradation. Besides, without a power model like R-50 to lead the optimization direction during self-supervised training, R-18 will not get good results as we said in the Section Introduction.
>
> **3. no theory is provided**
>
> We provide theoretical results to explain the working mechanism of slimmable models in linear
> cases. We also use such results to explain why a single slimmable proble layer does not work in practice.
> Please check Appendix A.1.
> Besides, tons of visualization results about why training slimmable networks is harder than that in supervised cases are provided in Appendix A.3.

---

### Official Review · Reviewer_yYC4 · 2022-11-03

**Confidence:** 3
**Clarity, Quality, Novelty And Reproducibility:** 1. The paper is well written and easy…
**Correctness:** 3
**Technical Novelty And Significance:** 2
**Empirical Novelty And Significance:** 2
**Recommendation:** 5

**Strength And Weaknesses:**

1. The proposed approach achieved balanced performance between multiple architecture sizes and outperformed the other small SSL models.
2. The paper is well written and easy to follow.

Weakness
1. The novelty is limited as this paper simply applies the slimmable architecture with some minor improvements on the training loss.
2. Given the paper target on training a small SSL model, it would be interesting to compare the performance with the small distilled model in terms of accuracy, model size, FLOPs. The current evaluation is not sufficient to justify the motivation.
3. The evaluation can be improved by evaluating with other downstream datasets and tasks.


**Summary Of The Paper:**

The paper proposed a slimmable contrastive self-supervised learning framework for building small models with SimCLR. It applied the well-known slimmable network and solved the gradient imbalance problem in the training by slow start training, online distillation, and loss reweighting, etc.


**Summary Of The Review:**

The paper studied an interesting topic and solved it by applying an existing approach with some minor improvements on the training loss. The evaluation is not sufficient to justify the motivation and contributions.

---

> ### Author Response · Authors · 2022-11-18
> **response**
>
> **1. novelty is limited**
>
> Please do not ignore the main contribution of the paper: introducing a new solution (slimmable networks) to get self-supervised small models, finding the problems during this process, and making it work perfectly in self-supervised learning. It is not so “simple”. It is new to introduce slimmable networks into self-supervised learning, and we are the first ones to do such a thing. It is also non-trivial to find the gradient imbalance and gradient direction divergence phenomenon of training slimmable networks in self-supervised learning (refer to Appendix A.3 VISUALIZATION). Borrowing the wisdom of previous works to solve new problems is not something wrong. We believe our work brings new findings to the community and may inspire some following works.
>
> Besides, we also make some theoretical contributions to explain the working mechanism of slimmable models in linear
> cases. Please check Appendix A.1.
>
>
> **2. compare the performance with the small distilled model in terms of accuracy, model size, FLOPs**
>
> The accuracy, model size, and FLOPs are all shown in Table 1 (the last two columns). With slimmable networks, we can get small models with better performance, smaller model sizes, and less FLOPs. Besides, you can also check the training time in Sec 4.3 slow start and training time. Specifically, The pre-training time of SlimCLR-MoCov2 without and with slow start epoch S = 100 on 8 Tesla V100 GPUs are 45 and 33 hours, respectively. For reference, the pre-training time of MoCov2 with ResNet-50 is roughly 20 hours. We only cost only ~50\% additional training time but get 4 different networks that are suitable for different devices.
>
> **3. The evaluation can be improved by evaluating with other downstream datasets and tasks**
>
> Please check the newly added Table 3 in the revision. We add transfer learning results for object detection and instance segmentation tasks. The pre-trained model of SlimCLR outperform the supervised baseline in these two tasks on COCO 2017.

---

### Author Response · Authors · 2022-12-10
**Any response to the rebuttal?**

Hi, is there any response to the rebuttal? We are happy to hear some noise.

---

### Decision · Program_Chairs · 2023-01-20

**Decision:**

Reject

**Justification For Why Not Higher Score:**

The present draft discusses a mixture of previously proposed building blocks in a context that is not properly motivated. The paper could be improved by improving the positioning and finding unique properties that using slimmable nets with SSL unlocks.

**Justification For Why Not Lower Score:**

N/A

**Metareview: Summary, Strengths And Weaknesses:**

This paper proposes to use slimmable networks for contrastive self-supervised learning. Then, it suggests using several tricks to improve the performance of sub-networks. While reviewers have highlighted a few positive points about the paper, the novelty and contributions of this work were questioned. The techniques presented in this paper are not new, and the draft describes a combination thereof. However, the full system does not unlock things that were not observed before or that would lead to drastic gains.
Some of the motivations of this work are also problematic, for instance regarding gradient imbalance and specificities of SSL (in the context of slimmable nets). Finally, the treatment of related work is not thorough enough.
For all those reasons, I recommend rejecting this paper.

**Summary Of Ac-Reviewer Meeting:**

N/A